# Commit to the Bit: Reactive Reinforcement Learning Done Right

**Onno Eberhard** [1 2] **Claire Vernade** [3] **Michael Muehlebach** [1]

## Abstract

Reinforcement learning algorithms are commonly analyzed (and designed) under the Markov assumption. This is unrealistic, as most environments encountered in practice are either partially observable, or require function approximation that restricts the agent to access non-Markovian state features. We consider the problem of learning an optimal reactive policy in a finite environment with deterministic observations (or equivalently, hard state aggregation). We introduce a new algorithm, *Committed Q-learning*, and prove almost-sure convergence to the optimal reactive policy under an intuitive assumption we call *rewire-robustness*. This assumption is strictly weaker than the $q_\star$-realizability condition used in prior work. Our algorithm is a variant of classical Q-learning in which the behavior policy commits to a single action upon entering a feature, and only resamples actions when the observed feature changes. A crucial part of our analysis is the introduction of *quasi-Markov* environments.

## 1. Introduction

The goal of reinforcement learning (RL; Sutton & Barto, 2018) is to enable an agent to autonomously learn to act optimally in an unknown environment. Most RL algorithms are based on principles from dynamic programming, and often involve estimating a *value function*. The optimal value function $v_\star$ represents the remaining sum of rewards in each state under the optimal policy. Given $v_\star$, the optimal action in any state can be found by a single-step look-ahead. This fundamental principle enables RL algorithms to solve challenging control problems, from playing games (Mnih et al., 2015; Silver et al., 2016; Vinyals et al., 2019) to robotics (OpenAI et al., 2020; Dürr et al., 2026). All these environ-

[1]Max Planck Institute for Intelligent Systems, Tübingen, Germany [2]University of Tübingen [3]University of Technology Nuremberg. Correspondence to: Onno Eberhard <oeberhard@tue.mpg.de>.

*Proceedings of the 43rd International Conference on Machine Learning*, Seoul, South Korea. PMLR 306, 2026. Copyright 2026 by the author(s).

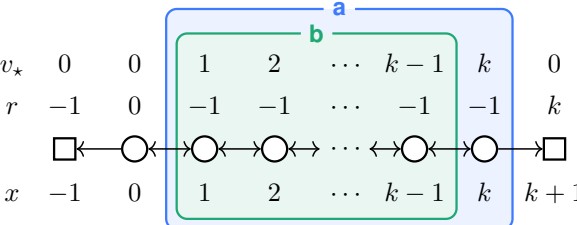

*Figure 1.* The corridor environment with states $x$, rewards $r$, and optimal value function $v_\star$. An episode begins in state $x = 0$ and ends after the agent enters either of the square states. **(a)** The blue bubble defines a state aggregation that turns the MDP into a *quasi-Markov* environment (Section 4). Here, the feature-value function can be clearly defined (Lemma 4.3). **(b)** The green bubble defines a state aggregation that is not quasi-Markov, but *rewire-robust* (Section 3). This property is sufficient for recovering the optimal reactive policy with Committed Q-learning (Theorem 3.5).

ments, like most that are relevant in practice, are either very large, such that the value function can only be approximated, or are partially observable. In both cases, the agent has to act with incomplete information about the environment's state, encoded in a *state feature*. Despite considerable progress, existing theory for this problem is still very limited (Bertsekas & Tsitsiklis, 1996; Agarwal et al., 2022). In this work, we focus on reinforcement learning in partially observable environments with deterministic observations. This setting is also known as *hard state aggregation*. While an optimal policy generally requires a memory of past observations (Kaelbling et al., 1998), we restrict ourselves to the problem of learning a *reactive* policy, i.e., one that does not depend on past features. This policy maps features to distributions over actions that we call *options* (Sutton et al., 1999), as deterministic policies may perform arbitrarily worse than stochastic policies (Singh et al., 1994b).

Consider the corridor environment, shown in Fig. 1. The optimal policy is to go right at every state, yielding a total sum of rewards of 0, which is preferable to the immediate termination with $-1$ on the left. Suppose that the agent cannot distinguish between the states in the blue bubble (Fig. 1a), making this environment partially observable. While this seems a trivial environment, classical value-based RL algorithms such as Q-learning (Watkins & Dayan, 1992) will fail if applied directly. The reason for this is that these methods try to approximate the optimal value function. However, the states underlying the blue feature have completely dif-

ferent values, hence the value of this feature is ambiguous. Existing theoretical results for this problem (Tsitsiklis & Van Roy, 1996; Majeed & Hutter, 2018) thus require $q_\star$-*realizability*, which means that the optimal value function should be constant inside each feature. In this paper, we show that this assumption is stronger than necessary in many cases, and that a slightly augmented version of Q-learning converges to the optimal reactive policy under a strictly weaker assumption: *rewire-robustness*.

Our key insight is that, despite state aggregation, there exists a value for the blue feature in Fig. 1a that enables dynamic programming to work: the value of the *entrance state*, in this case $v_\star(1)$. With this value, a look-ahead from $x = 0$ will prefer to enter the corridor, and a look-ahead from a corridor state will similarly prefer to go to the right. We call environments in which each feature has a unique entrance state (or a unique distribution over entrance states) *quasi-Markov*. Similar to the value function in a Markov decision process (MDP), the entrance values in quasi-Markov environments can be efficiently computed by dynamic programming. In particular, Q-learning converges to these values if the behavior policy only samples a new option when the observed feature changes. We call this augmented algorithm *Committed Q-learning* (Algorithm 1). The "commitment" to an option when entering a feature is not restrictive, since the goal of Q-learning is to find a reactive policy that deterministically maps features to options, and all such policies are naturally "committed."

Many environments are not quasi-Markov. For example, consider the green bubble in Fig. 1b. Here, there are two different entrance states ($x = 1$ and $x = k - 1$) with two different values. However, this environment is *rewire-robust*: if we change the connections into the corridor (for example, by switching the two entrance states), the optimal reactive policy is unaffected. We prove that any rewire-robust environment can be well approximated by a quasi-Markov environment and show that Committed Q-learning converges almost surely to the optimal reactive policy in all rewire-robust environments. We further show that this condition is strictly weaker than $q_\star$-realizability, which, to the best of our knowledge, is assumed in all prior work on this topic. Our convergence analysis is based on the recent extension of the Borkar-Meyn theorem (Borkar & Meyn, 2000) to Markovian noise by Liu et al. (2025).

## 2. Preliminaries

We consider the problem of reinforcement learning in a finite episodic partially observable Markov decision process (POMDP) with deterministic observations. The state space is of the form $\mathcal{X}' = \mathcal{X} \cup \{x^\perp\}$, where $\mathcal{X}$ is a finite set and $x^\perp$ is an absorbing terminal state. An episode begins at time $t = 0$ in a state $x_0 \in \mathcal{X}$ selected randomly according to the

initial state distribution $p_0 \in \Delta_{\mathcal{X}}$, such that $\mathbb{P}\{x_0 = x\} = p_0(x)$. At each time $t$ before termination, the environment is in a state $x_t \in \mathcal{X}$, and the agent selects an action $u_t$ from the finite action space $\mathcal{U}$. The environment then transitions to the next state $x_{t+1} \in \mathcal{X}'$ with probability

$$\mathbb{P}\{x_{t+1} = x' \mid x_t = x, u_t = u\} = (T'_u)_{x',x},$$

where $T'_u \in \Delta_{\mathcal{X}'}^{\mathcal{X}'}$ is the extended transition kernel under action $u$.[1] For nonterminal states $x$ and $x'$, this probability is given by $(T'_u)_{x',x} \doteq (T_u)_{x',x}$, where $T_u \in [0,1]^{\mathcal{X} \times \mathcal{X}}$ is the transition kernel of the environment. The termination probabilities are $(T'_u)_{x^\perp,x} \doteq 1 - \gamma_{x,u}$, where $\gamma_{x,u} \doteq \sum_{x' \in \mathcal{X}} (T_u)_{x',x}$. Finally, the terminal state $x^\perp$ is absorbing, and thus $(T'_u)_{x',x^\perp} = [x' = x^\perp]$, where $[\cdot]$ denotes the Iverson bracket.

A feature mapping (or observation function) $\varphi : \mathcal{X}' \to \mathcal{Z}'$ maps states to a finite feature space $\mathcal{Z}' = \mathcal{Z} \cup \{z^\perp\}$, where $\varphi(x) = z^\perp$ if and only if $x = x^\perp$. This mapping defines a partition of the state space as $\mathcal{X} = \dot\bigcup_z \mathcal{X}_z$, where $\mathcal{X}_z \doteq \{x \in \mathcal{X} \mid \varphi(x) = z\}$ for $z \in \mathcal{Z}$. It is often convenient to write $\varphi$ as a matrix $\Phi \in \{0,1\}^{\mathcal{Z} \times \mathcal{X}}$, where $\Phi_{z,x} \doteq \varphi_z(x) \doteq [\varphi(x) = z]$. At time $t$, the agent only observes the feature $z_t \doteq \varphi(x_t)$, which makes this environment partially observable. Given a reward function $r : \mathcal{Z}' \to \mathbb{R}$, with $r(z^\perp) \doteq 0$ and a finite set of options $\Omega \subset \Delta_{\mathcal{U}}$, we can define the value of a reactive policy $\pi : \mathcal{Z} \to \Omega$ in a state $x \in \mathcal{X}$ as the expected sum of rewards in an episode starting in $x$:

$$v_\pi(x) \doteq \mathbb{E}_\pi\left[\sum_{t=1}^{H} r(z_t) \mid x_0 = x\right],$$

where the notation $\mathbb{E}_\pi$ (or $\mathbb{P}_\pi$) indicates that $u_t \sim \pi(z_t)$, and where $H$ is the (random) termination time. Our goal is to learn a policy $\pi$ that maximizes $J(\pi) \doteq p_0^\top v_\pi$.

**Notation.** The following notation is used frequently throughout the article. The transition kernel under an option $\omega \in \Omega$ is denoted as $T_\omega \doteq \sum_{u \in \mathcal{U}} \omega(u) T_u$. Similarly, for $x \in \mathcal{X}$ and $\omega \in \Omega$, we write $\gamma_{x,\omega} \doteq \sum_{x' \in \mathcal{X}} (T_\omega)_{x',x}$. Given a feature $z \in \mathcal{Z}$, we define $\Pi_z \doteq \mathrm{diag}\,\varphi_z$ and $\Pi_z^\perp \doteq I - \Pi_z$. These are the projection matrices in $\mathbb{R}^{\mathcal{X}}$ onto the subspace corresponding to states with feature $z$ and onto the corresponding orthogonal subspace. Finally, an environment is denoted as $\mathcal{E} = (\mathcal{M}, \varphi)$, where $\mathcal{M} = (\{T_u\}, p_0, r)$ is the MDP underlying $\mathcal{E}$.

## 3. Main results

The Committed Q-learning algorithm is shown in Algorithm 1. Apart from the use of more general options instead of actions, the only change from the classical non-committed Q-learning algorithm is the if-statement in Line 9.

---

[1] This denotes that $(T'_u)_{:,x} \in \Delta_{\mathcal{X}'}$ for each $x \in \mathcal{X}'$ and $u \in \mathcal{U}$.

---

**Algorithm 1:** Committed Q-learning

**Input:** Environment $\mathcal{E}$, behavior policy $\pi : \mathcal{Z} \to \Delta_\Omega$,
step sizes $(\alpha_t) \subset \mathbb{R}$

1   $Q(z, \omega) \leftarrow 0$ for all $z \in \mathcal{Z}'$ and $\omega \in \Omega$
2   $x \sim p_0, \; z \leftarrow \varphi(x), \; \omega \sim \pi(z)$      // Start episode
3   **for** $t \in \mathbb{N}_0$ **do**
      // Sample transition
4      $u \sim \omega, \; x \sim (T'_u)_{:,x}, \; z' \leftarrow \varphi(x)$
      // Update Q-values
5      $\Delta \leftarrow r(z') + \max_{\omega' \in \Omega} Q(z', \omega') - Q(z, \omega)$
6      $Q(z, \omega) \leftarrow Q(z, \omega) + \alpha_t \Delta$
7      **if** $z' = z^\perp$ **then**      // Start new episode
8        $x \sim p_0, \; z \leftarrow \varphi(x), \; \omega \sim \pi(z)$
9      **else if** $z' \neq z$ *(or if non-committed)* **then**
10       $z \leftarrow z', \; \omega \sim \pi(z)$      // Sample new option

---

This change makes sure that a new option is only sampled if the observed feature changes. The algorithm iteratively updates a variable $Q \in \mathbb{R}^{\mathcal{Z}' \times \Omega}$. We denote the state of this variable at the beginning of the $t^{\text{th}}$ iteration (Line 3) as $Q_t$. Our convergence result relies on the following assumptions.

**Assumption 3.1.** All policies are *proper*, meaning that all episodes will eventually terminate. Formally, given any policy $\pi : \mathcal{Z} \to \Omega$ and any initial state $x \in \mathcal{X}$,

$$\mathbb{P}_\pi \{ \exists t : x_t = x^\perp \mid x_0 = x \} = 1. \qquad \Diamond$$

This is a very common assumption in the analysis of episodic reinforcement learning and dynamic programming algorithms (Bertsekas, 2012; Tsitsiklis, 1994; Jaakkola et al., 1993). It ensures that the Bellman operator of the episodic MDP $\mathcal{M}$ underlying the environment $\mathcal{E}$ is a contraction, which is essential for our convergence result.

**Assumption 3.2.** The behavior policy $\pi : \mathcal{Z} \to \Delta_\Omega$ satisfies $\pi(\omega \mid z) > 0$ for all $z \in \mathcal{Z}$ and $\omega \in \Omega$.    $\Diamond$

This assumption is clearly necessary for exploration.

**Assumption 3.3.** The step sizes $(\alpha_t) \subset \mathbb{R}$ are of the form $\alpha_t = \frac{\tau_1}{t + \tau_2}$ for some constants $\tau_1, \tau_2 > 0$.    $\Diamond$

This is a common assumption in the analysis of stochastic approximation algorithms of the form we study in this paper (Yu, 2012; Liu et al., 2025). In fact, the result of Liu et al. (2025) upon which our convergence analysis is based allows us to use a slightly more general form of step size, $\alpha_t = \frac{\tau_1}{(t + \tau_2)^\beta}$ with $\beta \in (0.5, 1]$, if we make the additional assumption that the Markov chain $(\xi_t)$ described in Section 5 is aperiodic.

Assumptions 3.1 to 3.3 suffice to establish almost-sure convergence of the sequence of iterates $(Q_t)$ in Algorithm 1 to a point $Q_\star$. However, additional assumptions are needed to guarantee optimality of the greedy policy with respect to

$Q_\star$. To see why, consider the environment shown in Fig. 4a (p. 8). The optimal policy moves right in state a and up in state b. A greedy policy selects the action based on the value $v_\star$ of the next feature. Writing $z \doteq \varphi(\mathtt{c}) = \varphi(\mathtt{d})$, we thus need that $v_\star(z) < 0$ (to go up in b) as well as that $v_\star(z) > v_\star(\mathtt{b})$ (to go right in a). However, since $v_\star(\mathtt{b}) = 0$, this is a contradiction. Thus, no single value $v_\star(z)$ is satisfactory. (The same argument works for Q-values as well.) To avoid situations like this, it is typically assumed that the optimal value function is *realizable* by the feature mapping $\varphi$ (e.g., Tsitsiklis & Van Roy, 1996; Majeed & Hutter, 2018). The formal definition of $q_\star$-realizability is the following.[2]

**Definition 3.4.** An environment $\mathcal{E} = (\mathcal{M}, \varphi)$ is called $q_\star$-*realizable* if there exists a function $q : \mathcal{Z} \times \mathcal{U} \to \mathbb{R}$ such that $q_\star(x, u) = q(\varphi(x), u)$ for all $x \in \mathcal{X}$ and $u \in \mathcal{U}$, where $q_\star$ is the optimal action-value function in $\mathcal{M}$.    $\Diamond$

If $\mathcal{E}$ is $q_\star$-realizable, then (non-committed) Q-learning converges to the optimal policy (Majeed & Hutter, 2018). However, $q_\star$-realizability is a very strong assumption that excludes simple environments like the corridor (Fig. 1a), for which we have already shown that an optimal greedy policy exists (see Section 1). In this work, we show that realizability is stronger than necessary, and that Committed Q-learning converges to the optimal reactive policy under the strictly weaker condition of *rewire-robustness*. We first state our main result, and then define rewire-robustness.

**Theorem 3.5.** *Let $\mathcal{E}$ be an environment and $\pi$ a behavior policy such that Assumptions 3.1 to 3.3 hold. Then, the iterates $(Q_t)$ of Algorithm 1 converge almost surely to a solution $Q_\star$, with corresponding greedy policy $\hat{\pi}_\star(z) \doteq \arg\max_{\omega \in \Omega} Q_\star(z, \omega)$, such that*

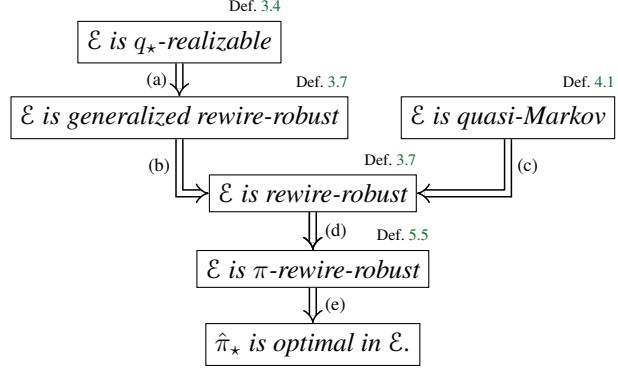

*Furthermore, if the greedy policy $\hat{\pi}_\star$ is unique, then* (e) *becomes an equivalence. All other implications are strict.*

*Proof.* See Section 5.    $\square$

**Rewire-robustness.** What is rewire-robustness? Consider the environment shown in Fig. 2a. There are two ways to enter into the aggregate feature: from state a (leading to c),

---

[2]Appendix E compares Definition 3.4 to (Weisz et al., 2021).

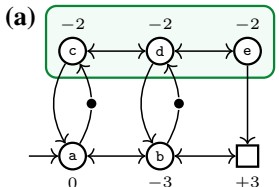 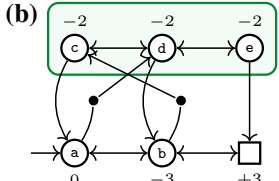 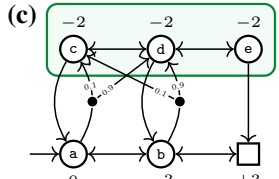 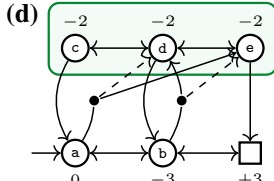

*Figure 2.* **(a)** A simple environment in which the states $\{c, d, e\}$ are aggregated into a single feature. For each state, the immediate reward is shown. The optimal policy is reactive and selects the direct path from $a$ via $b$ to the goal. The total reward of this path is 0. This environment is not $q_\star$-realizable, since the aggregated states have different optimal values. **(b)** A rewiring of the environment (a) changes the way a feature is entered, while keeping the set $\{c, d\}$ of possible entrance states fixed. Here, the environment is *rewire-robust*: the optimal policy in this rewiring is the same as in the original environment. **(c)** This rewiring of (a) is stochastic and *quasi-Markov* (see Section 4), since there is a unique entrance distribution. The $\pi$-*rewiring* introduced in Section 5 is of this type, where the probabilities are derived from the stationary distribution of the behavior policy $\pi$. **(d)** The solid lines represent a *generalized* rewiring of the environment (a): here, the set of entrance states $\{d, e\}$ is not a subset of $\{c, d\}$. In this environment, the optimal policy differs (from $a$ via $e$ to the goal) and achieves a higher total reward. Thus, the environment (a) is not generalized rewire-robust. Furthermore, the environment (d) is not rewire-robust: while the best policy achieves a return of $+1$, the rewiring indicated by the dashed lines admits a maximum return of 0.

and from state $b$ (leading to $d$). A *rewiring* of an environment $\mathcal{E}$ is a modified version of $\mathcal{E}$ in which these 'entrance dynamics' are changed, but the transition kernel remains otherwise unperturbed. For example, the environment shown in Fig. 2b is a rewiring of Fig. 2a. A stochastic behavior policy $\pi$ implicitly defines an 'average entrance' behavior for each feature. The $\pi$-*rewiring* of an environment replaces all entrances of a feature by this 'average entrance.' For example, if $\pi$ has low probability of moving up in state $a$ but high probability of moving up in state $b$, then the $\pi$-rewiring of Fig. 2a might look like Fig. 2c. While the entrance distributions in a rewiring are restricted to be mixtures of the entrance distributions of the original environment, in a *generalized* rewiring, a feature may be entered in an arbitrary way. Figure 2d shows a generalized rewiring of Fig. 2a. The formal definition of rewirings and rewire-robustness are as follows. (Quasi-Markov and $\pi$-rewire-robust environments will be defined in the following two sections.)

**Definition 3.6** (Rewiring). An environment $\bar{\mathcal{E}}$ with initial state distribution $\bar{p}_0$ and transition kernel $\{\bar{T}_u\}$ is a *rewiring* of an environment $\mathcal{E}$ with initial state distribution $p_0$ and transition kernel $\{T_u\}$ if, for all $u \in \mathcal{U}$ and $z \in \mathcal{Z}$, (i) $\Phi\bar{p}_0 = \Phi p_0$, (ii) $\Phi\bar{T}_u = \Phi T_u$, (iii) $\Pi_z\bar{T}_u\Pi_z = \Pi_z T_u\Pi_z$, and (iv) $\bar{\varsigma}_z \subset \varsigma_z$, where the *entrance space* $\varsigma_z$ is defined as

$$\varsigma_z \doteq \sum_{u \in \mathcal{U}} \text{range}(\Pi_z T_u \Pi_z^\perp) + \text{span}\{\Pi_z p_0\},$$

and $\bar{\varsigma}_z$ is defined analogously for $\bar{\mathcal{E}}$. If conditions (i) – (iii) hold, then $\bar{\mathcal{E}}$ is a *generalized rewiring*. ◇

**Definition 3.7** (Rewire-robustness). An environment $\mathcal{E}$ is called (generalized) *rewire-robust* if any optimal policy of any (generalized) rewiring of $\mathcal{E}$ is optimal in $\mathcal{E}$. ◇

The conditions (i) – (iv) of Definition 3.6 constrain how much the environment $\bar{\mathcal{E}}$ can differ from $\mathcal{E}$. Condition (i) ensures that, while the initial state distributions may be

different, the "initial feature distributions" must match:

$$\bar{p}_0(z) = \sum_{x \in \mathcal{X}_z} \bar{p}_0(x) = (\Phi\bar{p}_0)_z \stackrel{(i)}{=} (\Phi p_0)_z = p_0(z).$$

Similarly, condition (ii) ensures that $\bar{p}(z' \mid x, u) = p(z' \mid x, u)$. Condition (iii) demands that the intra-feature dynamics are unchanged: if $\varphi(x') = \varphi(x) \doteq z$, then

$$(\bar{T}_u)_{x',x} = (\Pi_z \bar{T}_u \Pi_z)_{x',x} \stackrel{(iii)}{=} (\Pi_z T_u \Pi_z)_{x',x} = (T_u)_{x',x}.$$

Finally, condition (iv) requires the feature entrance distributions of $\bar{\mathcal{E}}$ to be derived from $\mathcal{E}$. A feature $z$ can be entered either at the beginning of an episode, in which case the unnormalized entrance distribution over $\mathcal{X}_z$ is $\Pi_z p_0$, or it can be entered from a state $x$ outside of feature $z$, in which case the unnormalized entrance distribution is $(\Pi_z T_u \Pi_z^\perp)_{:,x}$. Thus, the entrance space $\varsigma_z$ is the linear span of all possible entrance distributions into feature $z$.

**Numerical example.** To motivate the relevance of Theorem 3.5, we illustrate the performance of Committed Q-learning in the corridor environment of Fig. 1a. The results are shown in Fig. 3. While non-committed Q-learning fails to make progress even with very short corridors, Committed Q-learning quickly converges to the optimal policy. In this experiment, the behavior policy at time $t$ is $\epsilon_t$-greedy with respect to the Q-table $Q_t$, where $\epsilon_t = \frac{a_\epsilon}{t + b_\epsilon}$ such that $\epsilon_0 = 0.1$ and $\epsilon_{1000} = 0.01$. Similarly, the step size is $\alpha_t = \frac{a_\alpha}{t + b_\alpha}$ such that $\alpha_0 = 0.1$ and $\alpha_{1000} = 0.01$. For the code, see https://github.com/onnoeberhard/q-commit.

## 4. Quasi-Markov environments

We now formalize the intuition developed in Section 1 that the correct value to assign to a feature is the average value of the feature's *entrance states*, and that this choice guarantees that the value is meaningful in a dynamic programming context. This observation is far from obvious. Indeed, in

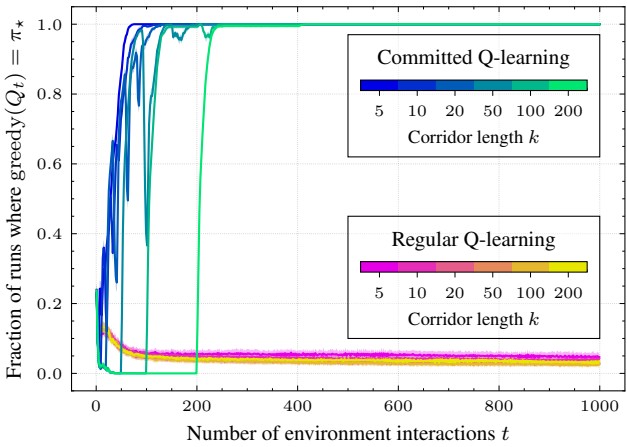

*Figure 3.* Learning curves of Committed Q-learning and regular Q-learning in corridor environments (Fig. 1a) of different lengths. The experiment is repeated with 1000 different random seeds, and we plot 95% bootstrap confidence intervals for the average optimality of the Q-table $Q_t$.

their original investigation of reactive policies under hard state aggregation, Singh et al. (1994b) propose that, given a policy $\pi : \mathcal{Z} \to \Omega$, the feature-value function $v : \mathcal{Z} \to \mathbb{R}$ should minimize the *value error*

$$\overline{\text{VE}}(v) \doteq \mathbb{E}_\mu\big[\{v(z_t) - \mathbb{E}_\pi[R_t \mid x_t]\}^2\big],$$

where $\mu$ is the stationary distribution of the policy $\pi$ in the environment $\mathcal{E}$, and $R_t$ is the sum of future rewards. The function $v$ that minimizes the value error is the $\mu$-weighted average of the values $v_\pi(x)$ of all states $x$ in a feature, where $v_\pi$ is the value function of the MDP $\mathcal{M}$ underlying $\mathcal{E}$. To see why this is not adequate, consider again the corridor environment from Fig. 1a, but suppose the reward of state $x = -1$ is $r(-1) = 2$, such that it is optimal to terminate immediately on the left. Under the always-right policy $\pi$, the stationary distribution is uniform in the corridor, and the values of the corridor states are $v_\pi(x) = x$. Thus, the corridor value $v_c$ that minimizes the value error is $\frac{1}{k}\sum_{x=1}^k v_\pi(x) = \frac{k+1}{2}$, which is greater than 2 if the corridor length $k$ is greater than 3, making the greedy policy suboptimal. Another common objective is the *Bellman error*

$$\overline{\text{BE}}(v) \doteq \mathbb{E}_\mu\big[\{v(z_t) - \mathbb{E}_\pi[r(z_{t+1}) + v(z_{t+1}) \mid x_t]\}^2\big].$$

Under the always-right policy in the corridor, the Bellman error is minimized by $v_c = k$ (Proposition B.3), which makes the greedy policy suboptimal for $k > 2$. Both the value error and the Bellman error include a conditioning on the states of $\mathcal{M}$ in their definitions, which leads to problems with *learnability* in the partially observable setting. This was first shown by Sutton & Barto (2018, Section 11.6), who demonstrate that it is not possible to estimate $\overline{\text{VE}}$ or $\overline{\text{BE}}$ from interaction with an environment. We thus propose the alternative definitions of *value risk*

$$\mathcal{R}_{\text{V}}(v) \doteq \mathbb{E}_\mu\big[\{v(z_t) - \mathbb{E}_\mu[R_t \mid z_t]\}^2\big]$$

and *Bellman risk*

$$\mathcal{R}_{\text{B}}(v) \doteq \mathbb{E}_\mu\big[\{v(z_t) - \mathbb{E}_\mu[r(z_{t+1}) + v(z_{t+1}) \mid z_t]\}^2\big].$$

These objectives do not have the same issues as $\overline{\text{VE}}$ or $\overline{\text{BE}}$ with learnability, as we show in Appendix D. Furthermore, returning to the corridor example, while minimizing the value risk is equivalent to minimizing the value error (Proposition D.2), minimizing the Bellman risk yields the desired solution. To see this, note that we can minimize the Bellman risk $\mathcal{R}_{\text{B}}$ by solving the Bellman equation

$$v(z) = \mathbb{E}_\mu[r(z_{t+1}) + v(z_{t+1}) \mid z_t = z]$$

for $v \in \mathbb{R}^{\mathcal{Z}}$. Solving for the value $v_c$ of the corridor state under the always-right policy, we get

$$v_c = \frac{k-1}{k}(-1 + v_c) + \frac{1}{k}(k + 0) \implies v_c = 1,$$

which is exactly the value of the corridor entrance state. In the rest of this section we prove that minimizing the Bellman risk will always result in a value function that satisfies this property, as long as the environment is *quasi-Markov*.

**Definition 4.1** (Quasi-Markov). An environment $\bar{\mathcal{E}}$ with initial state distribution $\bar{p}_0$ and transition kernel $\{\bar{T}_u\}$ is called *quasi-Markov* if there exists an *entrance matrix* $\Sigma \in \Delta_{\bar{\mathcal{X}}}^{\mathcal{Z}}$ with columns $\sigma_z$ satisfying $\sigma_z = \Pi_z \sigma_z$ and

$$\bar{p}_0 = \Sigma \Phi \bar{p}_0 \quad \text{and} \quad \Pi_z \bar{T}_u \Pi_z^\perp = \Pi_z \Sigma \Phi \bar{T}_u \Pi_z^\perp,$$

for all $z \in \mathcal{Z}$ and $u \in \mathcal{U}$. Equivalently, $\bar{\mathcal{E}}$ is quasi-Markov, if the entrance space satisfies $\dim \bar{\varsigma}_z \leq 1$ for all $z \in \mathcal{Z}$. $\diamond$

Thus, if an environment is quasi-Markov, then the state $x \in \mathcal{X}_z$ will be distributed according to $\sigma_z$ whenever the feature $z$ is entered from the outside, independent of what the previous state was: if $z \doteq \varphi(x_t) \neq \varphi(x_{t-1})$, then

$$\mathbb{P}\{x_t = x \mid x_{t-1} = x_-, u_{t-1} = u\}$$
$$= (\Pi_z \Sigma \Phi \bar{T}_u \Pi_z^\perp)_{x, x_-} = (\Pi_z \Sigma \zeta)_x = \zeta_z \sigma_z(x),$$

with $\zeta \doteq (\Phi \bar{T}_u \Pi_z^\perp)_{:, x_-}$. This means that the Markov property holds whenever the feature changes: if $z_{t-1} \neq z_t$, then $x_t \perp x_{t-1} \mid z_t$. This property is also related to *semi-Markov* environments (Puterman, 1994; Sutton et al., 1999), where the state $x$ is similarly not revealed at every step. The equivalence to an entrance space dimension $\leq 1$ is proved formally in Lemma B.1; the intuition is given in the discussion from Section 3. If an environment is quasi-Markov, we can define an associated *aggregate MDP* as follows.

**Definition 4.2** (Aggregate MDP). Given a quasi-Markov environment $\bar{\mathcal{E}}$ satisfying Assumption 3.1 with initial state distribution $\bar{p}_0$, transition kernel $\{\bar{T}_u\}$, and entrance matrix $\Sigma$, the corresponding *aggregate MDP* $\hat{\mathcal{M}}$ is an episodic MDP with state space $\mathcal{Z}$, action space $\Omega$, initial state distribution

$\hat{p}_0 \doteq \Phi \bar{p}_0$ and transition kernel $\hat{T}_\omega \doteq \Phi \bar{T}_\omega \Psi_\omega$ for all $\omega \in \Omega$. The *disaggregation matrix* $\Psi_\omega \in \Delta_{\mathcal{X}}^{\mathcal{Z}}$ has the columns

$$\psi_z^\omega \doteq \tilde{\psi}_z^\omega / \mathbf{1}^\top \tilde{\psi}_z^\omega, \quad \text{where} \quad \tilde{\psi}_z^\omega \doteq (I - \Pi_z \bar{T}_\omega)^{-1} \sigma_z. \; \diamond$$

Lemma B.4 proves that the matrix inverse in the definition above is well-defined, as the spectral radius satisfies $\rho(\Pi_z \bar{T}_\omega) < 1$. Note that the term 'aggregate MDP' is commonly used to describe systems of this type with general disaggregation matrices (e.g., Bertsekas, 2012, Section 6.5). The 'Bayesian' disaggregation probabilities $\{\psi_z^\omega\}$ that we define above correspond to the posterior distributions over states in a feature $z$ under an option $\omega$. To see this, note that when the feature $z$ is entered, the initial distribution over states is $\sigma_z$. After one transition under option $\omega$, the probability mass on states in $z$ is $\Pi_z \bar{T}_\omega \sigma_z$. Continuing recursively like this, we see that the unnormalized stationary distribution comes from the Neumann series

$$\sum_{\tau=0}^{\infty} (\Pi_z \bar{T}_\omega)^\tau \sigma_z = (I - \Pi_z \bar{T}_\omega)^{-1} \sigma_z = \tilde{\psi}_z^\omega,$$

which converges as $\rho(\Pi_z \bar{T}_\omega) < 1$ (e.g., Meyer, 2023, Theorem 4.11.2). In Lemma 5.4, we show that the transition kernel of $\hat{\mathcal{M}}$ is closely related to our definition of the Bellman risk, as $(\hat{T}_\omega)_{z',z} = \mu(z' \mid z, \omega)$, where $\mu$ is the stationary distribution of the behavior policy $\pi$ in the quasi-Markov environment $\bar{\mathcal{E}}$. Thus, the value function $\hat{v}_\pi$ of the aggregate MDP is the minimum Bellman risk solution in $\bar{\mathcal{E}}$. The following lemma shows that this solution will always satisfy the entrance value property described above.

**Lemma 4.3** (Entrance Value). *Let $\bar{\mathcal{E}} = (\bar{\mathcal{M}}, \varphi)$ be a quasi-Markov environment satisfying Assumption 3.1 with entrance matrix $\Sigma$ and let $\hat{\mathcal{M}}$ be the corresponding aggregate MDP. If $\pi : \mathcal{Z} \to \Omega$ is any policy, and $\bar{v}_\pi$ and $\hat{v}_\pi$ represent the value function of $\pi$ in $\bar{\mathcal{M}}$ and $\hat{\mathcal{M}}$, respectively, then*

$$\hat{v}_\pi = \Sigma^\top \bar{v}_\pi.$$

*Proof.* The aggregate value function $\hat{v}_\pi$ is the unique solution of the aggregate Bellman equation

$$\hat{v}_\pi(z) = \left\{ \hat{T}_{\pi(z)}^\top (r + \hat{v}_\pi) \right\}(z) \tag{1}$$

for all $z \in \mathcal{Z}$ (e.g., Bertsekas, 2012, Proposition 3.2.1). Our goal is thus to show that $\Sigma^\top \bar{v}_\pi$ solves this equation. The function $\bar{v}_\pi$ in turn is the unique solution of the Bellman equation in the MDP $\bar{\mathcal{M}}$,

$$\bar{v}_\pi(x) = \left\{ \bar{T}_{(\pi \circ \varphi)(x)}^\top (\Phi^\top r + \bar{v}_\pi) \right\}(x)$$

for all $x \in \mathcal{X}$. From now on, let $z \in \mathcal{Z}$ be fixed, and let $\omega \doteq \pi(z)$. From the definition of $\psi_z^\omega$, we have that

$$\sigma_z \propto (I - \Pi_z \bar{T}_\omega) \psi_z^\omega.$$

Since $\sigma_z$ is normalized, it follows that

$$\sigma_z = \frac{(I - \Pi_z \bar{T}_\omega) \psi_z^\omega}{\mathbf{1}^\top (I - \Pi_z \bar{T}_\omega) \psi_z^\omega} = \frac{(I - \Pi_z \bar{T}_\omega) \psi_z^\omega}{1 - \bar{\gamma}_{z,\omega}},$$

where $\bar{\gamma}_{z,\omega} \doteq \varphi_z^\top \bar{T}_\omega \psi_z^\omega$. Projecting $\bar{v}_\pi$ onto the subspace of $\mathbb{R}^{\mathcal{X}}$ corresponding to the feature $z$, we have

$$\Pi_z \bar{v}_\pi = \Pi_z \bar{T}_\omega^\top \Phi^\top r + \Pi_z \bar{T}_\omega^\top (\Pi_z + \Pi_z^\perp) \bar{v}_\pi.$$

Rearranging this equation, we get

$$\Pi_z \bar{v}_\pi = (I - \Pi_z \bar{T}_\omega)^{-\top} \Pi_z \bar{T}_\omega^\top (\Phi^\top r + \Pi_z^\perp \bar{v}_\pi).$$

This expression decomposes the value $\bar{v}_\pi(x)$ of states $x \in \mathcal{X}_z$ into a part that depends on the reward function, and a part that depends on the value of states *outside* of $\mathcal{X}_z$. Combining this with the above expression for $\sigma_z$, we have

$$\sigma_z^\top \bar{v}_\pi = \sigma_z^\top \Pi_z \bar{v}_\pi = \frac{(\Phi^\top r + \Pi_z^\perp \bar{v}_\pi)^\top \bar{T}_\omega \psi_z^\omega}{1 - \bar{\gamma}_{z,\omega}}.$$

We want to show that this expression solves (1). Plugging $\Sigma^\top \bar{v}_\pi$ into (1) and expanding, we get

$$
\begin{aligned}
&\left\{ \hat{T}_\omega^\top (r + \Sigma^\top \bar{v}_\pi) \right\}(z) \\
&= (\Phi \bar{T}_\omega \psi_z^\omega)^\top (r + \Sigma^\top \bar{v}_\pi) \\
&= (\bar{T}_\omega \psi_z^\omega)^\top \Phi^\top r + (\bar{T}_\omega \psi_z^\omega)^\top (\Pi_z + \Pi_z^\perp) \Phi^\top \Sigma^\top \bar{v}_\pi \\
&= (\Phi^\top r + \Pi_z^\perp \Phi^\top \Sigma^\top \bar{v}_\pi)^\top \bar{T}_\omega \psi_z^\omega + \bar{\gamma}_{z,\omega} \sigma_z^\top \bar{v}_\pi \\
&= (\Sigma^\top \bar{v}_\pi)(z) + \frac{\bar{\gamma}_{z,\omega}}{1 - \bar{\gamma}_{z,\omega}} (\Pi_z^\perp \bar{T}_\omega \psi_z^\omega)^\top (I - \Sigma \Phi)^\top \bar{v}_\pi.
\end{aligned}
$$

It remains to show that the second term vanishes. From the quasi-Markov property, we get, for any $z' \in \mathcal{Z}$ and $\psi \in \mathbb{R}^{\mathcal{X}}$,

$$\Pi_{z'} \bar{T}_\omega \Pi_{z'}^\perp \psi = \Pi_{z'} \Sigma \Phi \bar{T}_\omega \Pi_{z'}^\perp \psi = (\varphi_{z'}^\top \bar{T}_\omega \Pi_{z'}^\perp \psi) \sigma_{z'}.$$

This implies

$$\Pi_z^\perp \bar{T}_\omega \psi_z^\omega = \sum_{z' \neq z} \Pi_{z'} \bar{T}_\omega \Pi_{z'}^\perp \psi_z^\omega = \sum_{z' \neq z} \underbrace{(\varphi_{z'}^\top \bar{T}_\omega \psi_z^\omega)}_{(\hat{T}_\omega)_{z',z}} \sigma_{z'}.$$

Using the fact that $\varphi_z^\top \sigma_{z'} = [z = z']$ for all $z, z' \in \mathcal{Z}$, and thus $\Sigma \Phi \sigma_{z'} = \sigma_{z'}$, we get

$$(I - \Sigma \Phi)(\Pi_z^\perp \bar{T}_\omega \psi_z^\omega) = \sum_{z' \neq z} (\hat{T}_\omega)_{z',z} (I - \Sigma \Phi) \sigma_{z'} = 0.$$

Thus, we have shown that $\Sigma^\top \bar{v}_\pi$ is the unique solution of (1), and hence that $\hat{v}_\pi = \Sigma^\top \bar{v}_\pi$. $\qquad \square$

The following result shows why the entrance value property is important: it implies that an optimal policy in $\hat{\mathcal{M}}$ is an optimal reactive policy in $\bar{\mathcal{E}}$.

**Lemma 4.4.** *Let $\bar{\mathcal{E}}$ be a quasi-Markov environment satisfying Assumption 3.1 with corresponding aggregate MDP $\hat{\mathcal{M}}$, and let $\pi : \mathcal{Z} \to \Omega$ be any policy. Then, $\pi$ is optimal in $\bar{\mathcal{E}}$ if and only if it is optimal in $\hat{\mathcal{M}}$.*

*Proof.* This is a straightforward corollary of the previous lemma. The policy $\pi$ is optimal in $\bar{\mathcal{E}}$ if $\bar{J}(\pi) \geq \bar{J}(\pi')$ and optimal in $\hat{\mathcal{M}}$ if $\hat{J}(\pi) \geq \hat{J}(\pi')$ for all policies $\pi' : \mathcal{Z} \to \Omega$. From the Entrance Value Lemma (4.3) and the quasi-Markov property of $\bar{\mathcal{E}}$, we get, for any policy $\pi$

$$\hat{J}(\pi) \doteq \hat{p}_0^\top \hat{v}_\pi = (\Sigma \Phi \bar{p}_0)^\top \bar{v}_\pi = \bar{p}_0^\top \bar{v}_\pi \doteq \bar{J}(\pi),$$

which immediately yields the desired result. $\qquad\square$

## 5. Proof of Theorem 3.5

In this section, we use the theory of quasi-Markov environments developed above to prove Theorem 3.5. We focus on the convergence of Algorithm 1 and on the implication (e), which characterizes the optimality of the policy $\hat{\pi}_\star$. The remainder of Theorem 3.5 is straightforward and is discussed at the end of this section. The idea behind the proof is that, on average, the updates in Algorithm 1 are sampled from $\mu(z' \mid z, \omega)$, where $\mu$ is the stationary distribution induced by the behavior policy $\pi$. This distribution can be interpreted as the transition kernel of an aggregate MDP $\hat{\mathcal{M}}_\pi$ corresponding to a specific quasi-Markov environment $\bar{\mathcal{E}}_\pi$. We first prove that Algorithm 1 converges to the optimal action-value function in $\hat{\mathcal{M}}_\pi$. Then, Lemma 4.4 allows us to establish that $\hat{\pi}_\star$ is optimal in $\bar{\mathcal{E}}_\pi$. Finally, we use rewire-robustness to conclude that $\hat{\pi}_\star$ is optimal in $\mathcal{E}$.

Given an environment $\mathcal{E}$, a behavior policy $\pi$, and a sequence of step sizes $(\alpha_t)$, we define the random variables $x_t, \omega_t,$ and $Q_t$, for all $t \in \mathbb{N}_0$, as the states of the variables $x, \omega,$ and $Q$ at the beginning of the $t^{\text{th}}$ loop of Algorithm 1 (Line 3). We also define $x'_t$ as the state of $x$ immediately after Line 4, and we write $z_t \doteq \varphi(x_t)$ and $z'_t \doteq \varphi(x'_t)$. It can be verified by looking at Algorithm 1 that the tuple $\xi_t \doteq (x_t, \omega_t, x'_t)$ is a Markov chain with transition kernel

$$\mathbb{P}'\{\xi_t = (x, \omega, x') \mid \xi_{t-1} = (x_-, \omega_-, x'_-)\}$$
$$= p(x \mid x'_-) \, p(\omega \mid x_-, \omega_-, x'_-, x) \, (T'_\omega)_{x',x},$$

where, writing $z \doteq \varphi(x)$ (and similarly for $z_-$ and $z'_-$),

$$p(x \mid x'_-) = \begin{cases} p_0(x) & \text{if } z'_- = z^\perp \\ [x = x'_-] & \text{otherwise, and} \end{cases}$$

$$p(\omega \mid x_-, \omega_-, x'_-, x) = \begin{cases} \pi(\omega \mid z) & \text{if } z'_- \neq z_- \\ [\omega = \omega_-] & \text{otherwise.} \end{cases}$$

Note that $\mathbb{P}'$ is not the same as the transition kernel $\mathbb{P}$ described in Section 2, in which a transition to $x^\perp$ is absorbing,

as Line 8 of Algorithm 1 starts a new episode whenever the terminal state is reached. The Markov chain $(\xi_t)$ evolves in the set $\Xi \subset \mathcal{X} \times \Omega \times \mathcal{X}'$ of *reachable* tuples $\xi$. We first show that $(\xi_t)$ converges to a stationary distribution $\mu \in \Delta_\Xi$.

**Lemma 5.1.** *Let $\mathcal{E}$ be an environment and $\pi$ a behavior policy such that Assumptions 3.1 and 3.2 hold. Then, the Markov chain $(\xi_t) \subset \Xi$ is irreducible and has a unique stationary distribution $\mu \in \Delta_\Xi$ satisfying*

$$\mu(x, \omega, x') = \mu(x, \omega)(T'_\omega)_{x',x}$$

*for all $(x, \omega, x') \in \Xi$. Furthermore, this distribution satisfies, for all $z \in \mathcal{Z}$ and $\omega \in \Omega$,*

$$\Pi_z \mu_\omega = (1 - \gamma_\pi) \pi(\omega \mid z) \Pi_z p_0 + \Pi_z T_\omega \Pi_z \mu_\omega$$
$$+ \pi(\omega \mid z) \sum_{\omega' \in \Omega} \Pi_z T_{\omega'} \Pi_z^\perp \mu_{\omega'},$$

*where $\mu_\omega \in \mathbb{R}^\mathcal{X}$ is defined as $\mu_\omega(x) \doteq \mu(x, \omega)$, and where $\gamma_\pi \doteq \sum_{x \in \mathcal{X}, \omega \in \Omega} \mu(x, \omega) \gamma_{x, \omega}$.*

*Proof.* See Appendix C. $\qquad\square$

Note that $\mu(\xi) > 0$ for all $\xi \in \Xi$. For convenience, we extend $\mu$ to the space $\Delta_{\mathcal{X} \times \Omega \times \mathcal{X}'}$ and define $\mu(x, \omega, x') = 0$ for all $(x, \omega, x') \notin \Xi$. In the following, we assume (without loss of generality) that all features are reachable under $\pi$, and define $\mu(z, \omega, z') \doteq \sum_{x \in \mathcal{X}_z} \sum_{x' \in \mathcal{X}_{z'}} \mu(x, \omega, x')$ for $z, z' \in \mathcal{Z}$. We can now prove the convergence of Algorithm 1 and characterize the solution $Q_\star$. The partial observability in $\mathcal{E}$ leads to correlated noise in the Q-learning updates of Algorithm 1, which makes it difficult to apply standard stochastic approximation results that are typically used to analyze Q-learning. Our convergence result is instead based on the recent development by Liu et al. (2025) which extends the Borkar-Meyn theorem to Markovian noise.

**Lemma 5.2.** *Let $\mathcal{E}$ be an environment and $\pi$ a behavior policy such that Assumptions 3.1 to 3.3 hold. Then, the iterates $Q_t$ of Algorithm 1 converge almost surely to a solution $Q_\star$ satisfying*

$$Q_\star(z, \omega) = \mathbb{E}_\mu\big[r(z') + \max_{\omega' \in \Omega} Q_\star(z', \omega') \mid z, \omega\big]. \quad (2)$$

*Proof.* See Appendix C. $\qquad\square$

Thus, $Q_\star$ is the optimal action-value function in an MDP on $\mathcal{Z}$ with transition dynamics $\mu$. We now show that this MDP can be interpreted as an aggregate MDP.

**Definition 5.3** ($\pi$-rewiring $\bar{\mathcal{E}}_\pi$, $\pi$-MDP $\hat{\mathcal{M}}_\pi$)**.** Let $\mathcal{E}$ be an environment with initial state distribution $p_0$ and transition kernel $\{T_u\}$ and $\pi$ a behavior policy such that Assumptions 3.1 and 3.2 hold. Then, the $\pi$-*rewiring* of $\mathcal{E}$ is the environment $\bar{\mathcal{E}}_\pi$ with initial state distribution $\bar{p}_0 \doteq \Sigma \Phi p_0$ and transition kernel

$$(\bar{T}_u)_{x',x} \doteq \begin{cases} (T_u)_{x',x} & \text{if } \varphi(x') = \varphi(x) \\ (\Sigma \Phi T_u)_{x',x} & \text{otherwise} \end{cases}$$

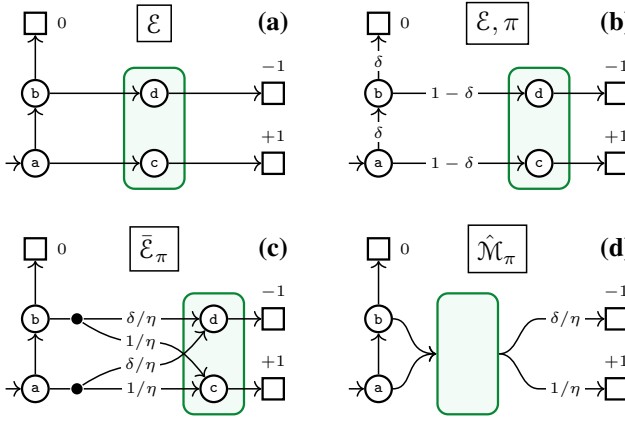

*Figure 4.* **(a)** A non-rewire-robust environment $\mathcal{E} = (\mathcal{M}, \varphi)$, where $\varphi(\mathtt{c}) = \varphi(\mathtt{d}) \doteq z$. No single value $v_\star(z)$ defines an optimal greedy policy. **(b)** A behavior policy $\pi$ is applied to the environment $\mathcal{E}$. **(c)** The $\pi$-rewiring $\bar{\mathcal{E}}_\pi$ of the environment $\mathcal{E}$ under the policy $\pi$ is a quasi-Markov rewiring of $\mathcal{E}$ in which the entrance distributions are derived from the stationary distribution $\mu$. **(d)** The $\pi$-MDP $\hat{\mathcal{M}}_\pi$ is the aggregate MDP corresponding to $\bar{\mathcal{E}}_\pi$.

for all $u \in \mathcal{U}$ and $x, x' \in \mathcal{X}$. The entrance matrix $\Sigma \in \Delta_{\mathcal{X}}^{\mathcal{Z}}$ is defined by the columns $\sigma_z \doteq \tilde{\sigma}_z / \mathbf{1}^\top \tilde{\sigma}_z$ with

$$\tilde{\sigma}_z \doteq \sum_{\omega \in \Omega} \Pi_z T_\omega \Pi_z^\perp \mu_\omega + (1 - \gamma_\pi) \Pi_z p_0.$$

Furthermore, the $\pi$-*MDP* $\hat{\mathcal{M}}_\pi$ associated with $\mathcal{E}$ is defined as the aggregate MDP corresponding to $\bar{\mathcal{E}}_\pi$. ◇

Lemma B.2 proves that $\bar{\mathcal{E}}_\pi$ is a quasi-Markov rewiring of $\mathcal{E}$ satisfying Assumption 3.1, such that $\hat{\mathcal{M}}_\pi$ is well-defined. The entrance distribution $\sigma_z$ is chosen to reflect the 'average' entrance distribution into $z$ under the behavior policy $\pi$. Consider again the environment shown in Fig. 4a. Suppose the behavior policy $\pi$ moves up with probability $\delta$ and right with probability $1 - \delta$ (Fig. 4b). In this case, the green feature is entered at state c with probability $1 - \delta$ (from a) and at state d with probability $\delta(1 - \delta)$ (from b). We thus see that the $\pi$-rewiring is given by Fig. 4c, where $\eta \doteq 1 + \delta$ is the normalizing constant. Finally, the $\pi$-MDP is given by Fig. 4d, as the distribution over the states in the green feature of $\bar{\mathcal{E}}_\pi$ is given by $\psi(\mathtt{c}) = 1/\eta$ and $\psi(\mathtt{d}) = \delta/\eta$. The following lemma provides the bridge between our convergence result (Lemma 5.2) and the quasi-Markov theory developed in Section 4.

**Lemma 5.4.** *Let $\mathcal{E}$ be an environment and $\pi$ a behavior policy such that Assumptions 3.1 and 3.2 hold. Then, if $\hat{\mathcal{M}}_\pi$ is the corresponding $\pi$-MDP with transition kernel $\{\hat{T}_\omega\}$,*

$$\mu(z' \mid z, \omega) = (\hat{T}_\omega)_{z', z}$$

*for all $z, z' \in \mathcal{Z}$, and all $\omega \in \Omega$.*

*Proof.* See Appendix C. □

Thus, (2) is in fact the Bellman optimality equation of the $\pi$-MDP $\hat{\mathcal{M}}_\pi$. This means that $\hat{\pi}_\star$ is an optimal policy in $\hat{\mathcal{M}}_\pi$, and, by Lemma 4.4, it is an optimal reactive policy in $\bar{\mathcal{E}}_\pi$. To guarantee that $\hat{\pi}_\star$ is also optimal in $\mathcal{E}$, we need the following assumption.

**Definition 5.5** ($\pi$-rewire-robustness). Let $\mathcal{E}$ be an environment and $\pi$ a behavior policy such that Assumptions 3.1 and 3.2 hold. Then, $\mathcal{E}$ is $\pi$-*rewire-robust* if any optimal reactive policy in $\bar{\mathcal{E}}_\pi$ is also optimal in $\mathcal{E}$. ◇

Thus, we have proved (e) of Theorem 3.5. Note that if $\hat{\pi}_\star$ is the unique minimizer of $Q_\star$, then the inverse direction of (e) follows immediately, since $\hat{\pi}_\star$ is the only optimal policy of $\hat{\mathcal{M}}_\pi$ and, by Lemma 4.4, of $\bar{\mathcal{E}}_\pi$. The implications (b) and (d) of Theorem 3.5 are trivial, since any rewiring is a generalized rewiring by definition, and we have shown that the $\pi$-rewiring is a rewiring (Lemma B.2). Implication (a) is proved in Proposition E.1 and (c) is proved in Lemma B.5. Furthermore (a) is strict, as the corridor environment (Fig. 1a) is generalized rewire-robust, but not $q_\star$-realizable. Implications (b) and (c) are strict, as the environment shown in Fig. 2a is rewire-robust, but neither generalized rewire-robust, nor quasi-Markov. Finally, the strictness of (d) is proved in Lemma B.6.

## 6. Related work

The concept of state aggregation goes back to the early days of dynamic programming (Whitt, 1978) and has been a part of reinforcement learning since the field's inception (Barto et al., 1983). Soon after the initial convergence results for Q-learning based on stochastic approximation theory were established (Tsitsiklis, 1994; Jaakkola et al., 1993), similar results were proved for reinforcement learning with state aggregation (Tsitsiklis & Van Roy, 1996; Singh et al., 1994a;b; Gordon, 1995). More recently, the theory of state aggregation has been extended by several researchers (Van Roy, 2006; Hutter, 2016; Majeed & Hutter, 2018; Dong et al., 2019). All these results either require $q_\star$-realizability to guarantee convergence to an optimal reactive policy, or they simply prove convergence of the algorithm without an interpretation of the solution. An assumption about the environment is likely necessary to enable an efficient algorithm, as Littman (1994) showed that the problem of finding a reactive optimal policy under state aggregation is NP-complete. To the best of our knowledge, the *rewire-robustness* that we propose is the weakest assumption under which convergence to the optimal reactive policy has been proved.

The *aggregate MDP* that we define in Section 4 is a classic tool for solving dynamic programming problems with state aggregation (Bertsekas, 2012, Section 6.5). However, there is little discussion on how to choose the disaggregation probabilities $\psi_z(x)$, since under $q_\star$-realizability, solving the aggregate MDP with *any* disaggregation matrix $\Psi$ will yield the optimal value function. To the best of our knowledge,

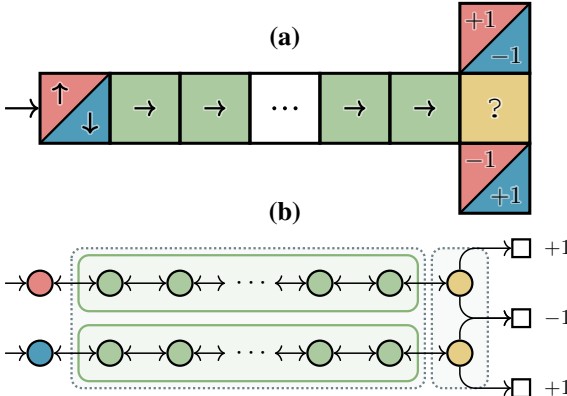

*Figure 5.* **(a)** The T-maze environment is a classic problem in partially observable RL. **(b)** The POMDP representation of the T-maze. The memory structure $z_t \doteq (y_0, y_t)$ defines a rewire-robust state aggregation of the underlying MDP, shown with green bubbles. The dotted gray bubbles show what the state aggregation would look like without memory. This state aggregation is not rewire-robust, and a reactive policy is not sufficient.

the work of Van Roy (2006) is the only study in which the disaggregation probabilities are chosen as in this paper, namely as the Bayesian posterior probabilities under the stationary distribution of $\pi$. However, since Van Roy considers general environments, not quasi-Markov environments, the posterior probabilities $\psi_z(x)$ depend on the policy $\pi$ (and not just on the action inside feature $z$ as for us). This makes it difficult to use this choice of disaggregation in practice, and indeed, while Van Roy studies the projected Bellman equation associated with this aggregate MDP, and shows that the Bayesian disaggregation matrix enables a better performance loss bound for value iteration, he does not propose a concrete algorithm for solving the projected Bellman equation. In fact, a few years later, Bertsekas (2011, p. 327) writes *"It is not clear whether it is practically advantageous to select [Ψ] in the manner suggested by Van Roy."*

The corridor environment (Fig. 1) is an augmented version of the T-maze (Bakker, 2001), which is a classic example in the literature on partially observable RL (Eberhard et al., 2025a; Allen et al., 2024). The T-maze is shown in Fig. 5a. The agent receives an observation of either 'up' or 'down' when entering the maze from the left. It then enters a corridor before reaching a decision point '?', where the optimal action (up or down) is determined by the initial observation. In this environment, a reactive policy is clearly not sufficient, and a memory of past observations is needed. The optimal memory is $z_t = (y_0, y_t)$, which contains the first and current observations. In Fig. 5b, the POMDP representation of this environment is shown, and it can be seen that this choice of memory is a state aggregation of the underlying MDP. Furthermore, this environment is clearly rewire-robust, for the same reason as the corridor environment. Thus, given this memory mechanism, Committed Q-learning will find the optimal policy mapping memory to actions.

## 7. Conclusion

We have introduced *Committed Q-learning*, an algorithm which provably converges to the optimal reactive policy in POMDPs with deterministic observations under an intuitive *rewire-robustness* condition, which is strictly weaker than the common $q_\star$-realizability assumption. Along the way, we have introduced the concepts of *Bellman risk* and *quasi-Markov* environments, which are natural extensions of prior work, and are crucial for our analysis of Committed Q-learning. In a quasi-Markov environment, the Markov property holds *between the features*, since the state $x$ is always independent of all previous states before entering the feature, but not *inside a feature*. While Committed Q-learning shares many similarities with regular Q-learning on the surface, in Section 5 we showed that the former effectively operates on a higher level: the $\pi$-MDP, whose state space is the feature space. This is reminiscent of the junction tree algorithm for inference in graphical models (e.g., Murphy, 2023, Section 9.6), where efficient message passing operations are performed between cliques, while a brute-force computation is performed inside each clique. In our case, the brute-force computation is the maximization step in Line 5 of Algorithm 1. We assume the existence of a finite set of options (distributions over actions) to optimize over. However, it is not clear how large such a set normally needs to be, as Vlassis et al. (2012) proved that such an optimization is, in general, NP-hard.

There are several open avenues for future research. First, while we consider stochastic options instead of deterministic actions, Singh et al. (1994b) also showed that nonstationary options are preferable to stationary options. It would thus be interesting to see whether our theory can be extended to temporally extended options (Sutton et al., 1999). This is a natural extension, since our algorithm already commits to a single option that is kept until a feature is exited. Optimizing over nonstationary options is an open-loop reinforcement learning problem (Eberhard et al., 2025b). Another interesting direction would be to extend our result to other types of RL problems than the episodic environments (stochastic shortest path problems) that we consider here. We expect that the analysis transfers with minor modification to discounted infinite-horizon problems, but it is less clear if it also extends to the average reward setting. Finally, while the rewire-robustness assumption is weaker than $q_\star$-realizability, it is less clear how to extend this theory to cases where the assumption *approximately* holds. In previous results on state aggregation (Tsitsiklis & Van Roy, 1996; Van Roy, 2006; Dong et al., 2019), the performance degrades with the approximation error on $q_\star$. It would be important to understand if there is a natural definition of 'approximate' rewire-robustness under which a similar line of analysis can be established.

## Acknowledgments

We thank the International Max Planck Research School for Intelligent Systems (IMPRS-IS) for their support. C. Vernade is funded by the Deutsch Forschungsgemeinschaft (DFG, German Research Foundation) under both the project 468806714 of the Emmy Noether Programme and under Germany's Excellence Strategy – EXC number 2064/1 – Project number 390727645, and also gratefully acknowledges funding from the European Union (ERC grant ConSequentIAL, number 101165883). M. Muehlebach is funded by the DFG under the project 456587626 of the Emmy Noether Programme. Views and opinions expressed are those of the authors only and do not necessarily reflect those of the European Union or the European Research Council. Neither the European Union nor the granting authority can be held responsible for them.

## Impact statement

This paper presents work whose goal is to advance the field of machine learning. There are many potential societal consequences of our work, none of which we feel must be specifically highlighted here.

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

# Appendix

## Contents

## A. Auxiliary results

**Theorem A.1** (Liu et al., 2025). *Given an initial vector $\theta_0 \in \mathbb{R}^d$ and a fixed sequence of step sizes $(\alpha_t) \subset \mathbb{R}$, consider the stochastic approximation recursion*

$$\theta_{t+1} = \theta_t + \alpha_t H(\theta_t, \xi_t), \tag{3}$$

*where $(\xi_t) \subset \Xi$ is a sequence of random variables following a Markov chain in a finite state space $\Xi$ and $H : \mathbb{R}^d \times \Xi \to \mathbb{R}^d$ is measurable. Assume the following.*

1. *The Markov chain $(\xi_t)$ is irreducible and thus has a unique stationary distribution $\mu \in \Delta_\Xi$.*

2. *The step sizes are of the form $\alpha_t = \frac{\tau_1}{t+\tau_2}$ for some constants $\tau_1, \tau_2 > 0$.*

3. *There exist measurable functions $H_\infty : \mathbb{R}^d \times \Xi \to \mathbb{R}^d$ and $b : \Xi \to \mathbb{R}^d$ such that, for any $\theta \in \mathbb{R}^d$, $\xi \in \Xi$, and $c \geq 1$,*
$$H(c\theta, \xi) - cH_\infty(\theta, \xi) = b(\xi).$$

4. *There exists a Lipschitz constant $L \geq 0$ such that, for any $\theta, \theta' \in \mathbb{R}^d$ and $\xi \in \Xi$,*
$$\|H(\theta, \xi) - H(\theta', \xi)\| \leq L\|\theta - \theta'\| \ \text{and}$$
$$\|H_\infty(\theta, \xi) - H_\infty(\theta', \xi)\| \leq L\|\theta - \theta'\|,$$
*where $\|\cdot\|$ denotes the maximum norm. Moreover, both*
$$h(\theta) \doteq \mathbb{E}_\mu H(\theta, \xi) \quad \text{and} \quad h_\infty(\theta) \doteq \mathbb{E}_\mu H_\infty(\theta, \xi)$$
*are well-defined and finite.*

5. *Let $h_c(\theta) \doteq h(c\theta)/c$ for all $\theta \in \mathbb{R}^d$ and $c \geq 1$. Then, $h_c \to h_\infty$ uniformly on any compact subset of $\mathbb{R}^d$ as $c \to \infty$. The ordinary differential equation (ODE)*
$$\dot{\theta}(t) = h_\infty\big(\theta(t)\big)$$
*has 0 as its globally asymptotically stable equilibrium.*

*Then, if $\theta_\star \in \mathbb{R}^d$ is the globally asymptotically stable equilibrium of the ODE*

$$\dot{\theta}(t) = h\big(\theta(t)\big),$$

*the iterates $\theta_t$ of (3) converge almost surely to $\theta_\star$.*

*Proof.* This is a special case of Corollary 8 of (Liu et al., 2025), which is an extension of the Borkar-Meyn theorem (Borkar & Meyn, 2000) to the case of Markovian noise. □

**Theorem A.2** (Borkar & Soumyanatha, 1997). *Let $F$ be a max-norm nonexpansive operator on $\mathbb{R}^d$ with a unique fixed point $\theta_\star \in \mathbb{R}^d$. Then, $\theta_\star$ is a globally asymptotically stable equilibrium point of the ODE*

$$\dot{\theta} = F\theta - \theta.$$

*Proof.* This is a special case of Theorem 3.1 of (Borkar & Soumyanatha, 1997). □

## B. Technical lemmas

**Lemma B.1.** *An environment $\mathcal{E}$ with initial state distribution $p_0$ and transition kernel $\{T_u\}$ is* quasi-Markov *if and only if the entrance space satisfies $\dim \varsigma_z \leq 1$ for all $z \in \mathcal{Z}$.*

*Proof.* First, suppose that $\mathcal{E}$ is quasi-Markov with entrance matrix $\Sigma$. The entrance space $\varsigma_z$ of feature $z$ is defined as

$$
\begin{aligned}
\varsigma_z &\doteq \sum_{u \in \mathcal{U}} \mathrm{range}(\Pi_z T_u \Pi_z^\perp) + \mathrm{span}\{\Pi_z p_0\} \\
&= \sum_{u \in \mathcal{U}} \mathrm{range}(\Pi_z \Sigma \Phi T_u \Pi_z^\perp) + \mathrm{span}\{\Pi_z \Sigma \Phi p_0\} \\
&\subset \mathrm{span}\{\sigma_z\},
\end{aligned}
$$

where we have used the quasi-Markov property. We thus have that $\dim \varsigma_z \leq 1$. For the other direction, let $z \in \mathcal{Z}$, and suppose that $\dim \varsigma_z \leq 1$. If $\dim \varsigma_z = 0$, then $\Pi_z p_0 = 0$ and $\Pi_z T_u \Pi_z^\perp = 0$ for all $u \in \mathcal{U}$. In this case, we can define $\sigma_z \doteq \frac{1}{|\mathcal{X}_z|}\varphi_z$. Otherwise, if $\dim \varsigma_z = 1$, we can define $\sigma_z$ as the unique vector $\sigma_z \in \varsigma_z \cap \Delta_\mathcal{X}$. In this way, we can build the entrance matrix $\Sigma$. We now verify that $\mathcal{E}$ is quasi-Markov with this entrance matrix. Let $z \in \mathcal{Z}$. First, suppose that $\dim \varsigma_z = 0$. Then,

$$\Pi_z \Sigma \Phi p_0 = \sigma_z \varphi_z^\top \Pi_z p_0 = 0 = \Pi_z p_0$$

and, for any $u \in \mathcal{U}$ and $x \in \mathcal{X}$,

$$
\begin{aligned}
(\Pi_z \Sigma \Phi T_u \Pi_z^\perp)_{:,x} &\\
= \sigma_z \varphi_z^\top (\Pi_z T_u \Pi_z^\perp)_{:,x} &= 0 = (\Pi_z T_u \Pi_z^\perp)_{:,x}.
\end{aligned}
$$

Now, suppose that $\dim \varsigma_z = 1$, and let $\sigma_z \in \varsigma_z \cap \Delta_\mathcal{X}$ be the unique vector described above. We know that $\Pi_z p_0 \in \varsigma_z$ and that $(\Pi_z T_u \Pi_z^\perp)_{:,x} \in \varsigma_z$, which, as $\sigma_z$ is normalized, implies that

$$\Pi_z p_0 = \sigma_z(\mathbf{1}^\top \Pi_z p_0) = (\varphi_z^\top p_0)\sigma_z = \Pi_z \Sigma \Phi p_0,$$

and similarly, that

$$(\Pi_z T_u \Pi_z^\perp)_{:,x} = \sigma_z \mathbf{1}^\top (\Pi_z T_u \Pi_z^\perp)_{:,x}$$
$$= \varphi_z^\top (T_u \Pi_z^\perp)_{:,x} \sigma_z$$
$$= (\Pi_z \Sigma \Phi T_u \Pi_z^\perp)_{:,x},$$

which concludes the proof. $\qquad\square$

**Lemma B.2.** *The $\pi$-rewiring $\bar{\mathcal{E}}_\pi$ of an environment $\mathcal{E}$ is a quasi-Markov rewiring of $\mathcal{E}$. Furthermore, $\bar{\mathcal{E}}_\pi$ satisfies Assumption 3.1.*

*Proof.* To show that $\bar{\mathcal{E}}_\pi$ is a rewiring, we have to verify the properties (i) – (iv) of Definition 3.6. Using the fact that $\varphi_z^\top \sigma_{z'} = [z = z']$ for all $z, z' \in \mathcal{Z}$, we get

$$(\Phi \Sigma \Phi)_{z,x} = \varphi_z^\top \sum_{z' \in \mathcal{Z}} \sigma_{z'} \Phi_{z',x} = \Phi_{z,x}$$

for all $z \in \mathcal{Z}$ and $x \in \mathcal{X}$. Thus, $\Phi \bar{p}_0 = \Phi \Sigma \Phi p_0 = \Phi p_0$ and $\Phi \bar{T}_u = \Phi T_u$, proving (i) and (ii). Condition (iii) is trivially fulfilled by definition of $\bar{T}_u$. For condition (iv), we need to show that $\bar{\varsigma}_z \subset \varsigma_z$. Let $\sigma \in \bar{\varsigma}_z$. Then,

$$\sigma = \sum_{u \in \mathcal{U}} \sum_{x \in \mathcal{X}} a_{u,x} (\Pi_z \bar{T}_u \Pi_z^\perp)_{:,x} + a_0 \Pi_z \bar{p}_0$$
$$= \sum_{u \in \mathcal{U}} \sum_{x \in \mathcal{X}} a_{u,x} (\Pi_z \Sigma \Phi T_u \Pi_z^\perp)_{:,x} + a_0 \Pi_z \Sigma \Phi p_0$$
$$\propto \sigma_z \propto \sum_{\omega \in \Omega} \Pi_z T_\omega \Pi_z^\perp \mu_\omega + (1 - \gamma_\pi) \Pi_z p_0 \subset \varsigma_z,$$

for some coefficients $\{a_{x,u}\}$ and $a_0$, from which we also get that $\sigma \subset \varsigma_z$, proving (iv). That $\bar{\mathcal{E}}_\pi$ is quasi-Markov follows immediately from the definition of $\bar{\mathcal{E}}_\pi$ and the rewiring property:

$$\bar{p}_0 = \Sigma \Phi p_0 = \Sigma \Phi \bar{p}_0,$$
$$\Pi_z \bar{T}_u \Pi_z^\perp = \Pi_z \Sigma \Phi T_u \Pi_z^\perp = \Pi_z \Sigma \Phi \bar{T}_u \Pi_z^\perp.$$

It remains to be shown that $\bar{\mathcal{E}}_\pi$ satisfies Assumption 3.1. First, note that a policy is proper, i.e., it eventually reaches the terminal state $x^\perp$ with probability 1 regardless of the starting state $x \in \mathcal{X}$, if and only if it has a positive probability of reaching $x^\perp$ in $|\mathcal{X}|$ steps regardless of the starting state $x \in \mathcal{X}$ (Bertsekas, 2012, Section 3.1). Let $\pi : \mathcal{Z} \to \Omega$ be any policy and let $x \in \mathcal{X}$. By Assumption 3.1, $\pi$ is proper in $\mathcal{E}$, which implies that there exists a $t \leq |\mathcal{X}|$ such that

$$\mathbb{P}_\pi\{x_t = x^\perp \mid x_0 = x\} > 0.$$

Thus, there exists a sequence of states $(\bar{x}_0, \bar{x}_1, \ldots, \bar{x}_t)$ and actions $(\bar{u}_0, \bar{u}_1, \ldots, \bar{u}_{t-1})$, with $\bar{x}_0 = x$ and $\bar{x}_t = x^\perp$, such that, for all $\tau \in [t]$, $(T_{\bar{u}_\tau})_{\bar{x}_{\tau+1}, \bar{x}_\tau} > 0$ and $\pi(\bar{x}_\tau)_{\bar{u}_\tau} > 0$. Denoting the probability measure of $\bar{\mathcal{E}}_\pi$ as $\bar{\mathbb{P}}$, we can show that $\pi$ is proper in $\bar{\mathcal{E}}_\pi$ by proving

$$\bar{\mathbb{P}}_\pi\{x_t = x^\perp \mid x_0 = x\} > 0,$$

as $x$ is arbitrary. We have that

$$\bar{\mathbb{P}}_\pi\{x_t = x^\perp \mid x_0 = x\}$$
$$\geq \bar{\mathbb{P}}_\pi\{\forall \tau \in [t] : x_\tau = \bar{x}_\tau, u_\tau = \bar{u}_\tau \mid x_0 = \bar{x}_0\}$$
$$= \prod_{\tau=0}^{t} \pi(\bar{x}_\tau)_{\bar{u}_\tau} (\bar{T}_{\bar{u}_\tau})_{\bar{x}_{\tau+1}, \bar{x}_\tau}.$$

We already know that the policy factors satisfy $\pi(\bar{x}_\tau)_{\bar{u}_\tau} > 0$ for all $\tau$. Furthermore, for all times $\tau$ where $\varphi(\bar{x}_{\tau+1}) = \varphi(\bar{x}_\tau)$, we have that $(\bar{T}_{\bar{u}_\tau})_{\bar{x}_{\tau+1}, \bar{x}_\tau} = (T_{\bar{u}_\tau})_{\bar{x}_{\tau+1}, \bar{x}_\tau} > 0$, as the intra-feature dynamics are unchanged. For the case where $z \doteq \varphi(\bar{x}_{\tau+1}) \neq \varphi(\bar{x}_\tau)$, we have, with $x \doteq \bar{x}_\tau$, $u \doteq \bar{u}_\tau$, and $x' \doteq \bar{x}_{\tau+1}$, that

$$(\bar{T}_u)_{x',x} = (\Sigma \Phi T_u)_{x',x} = \sigma_z(x') \underbrace{\varphi_z^\top (T_u)_{:,x}}_{>0},$$

where the second factor is positive since we know that a transition from $x$ to $x' \in \mathcal{X}_z$ is possible under $T_u$. The first factor, $\sigma_z(x')$ is positive if the unnormalized entrance probability $\tilde{\sigma}_z(x')$ is positive. From the definition of $\tilde{\sigma}_z$, we have that

$$\tilde{\sigma}_z(x') \geq \pi(x)_u (T_u)_{x,x_-} \mu(x, \pi(x)).$$

In this expression, all three factors are positive if we assume that $x$ is a reachable state. Note that, as in definition Definition 5.3, we can make this assumption without loss of generality, since unreachable states cannot possibly influence the convergence of Algorithm 1. Thus, we can conclude that $\bar{\mathcal{E}}_\pi$ indeed satisfies Assumption 3.1. $\qquad\square$

**Proposition B.3.** *Consider the corridor environment shown in Fig. 1a. The Bellman error $\overline{\mathrm{BE}}$ under the always-right policy $\pi$ is minimized with a corrdior value of $v_c = k$.*

*Proof.* If $\mu \in \Delta_\mathcal{X}$ is the stationary distribution of $\pi$ in the corridor environment, the Bellman error is

$$\overline{\mathrm{BE}}(v) = \sum_{x \in \mathcal{X}} \mu(x) \big\{ (v \circ \varphi)(x) - \mathbb{E}_\pi[r(z_+) + v(z_+) \mid x] \big\}^2.$$

Under the always-right policy, $\mu(x) = \frac{1}{k+1}$ for all states $x \in \mathcal{X}$ (excluding the terminal states). Thus, denoting the value of the initial state by $v_0$ and the value of the corridor feature by $v_c$,

$$\overline{\mathrm{BE}}(v) = \frac{1}{k+1} \{v_0 - (-1 + v_c)\}^2$$
$$+ \frac{k-1}{k+1} \{v_c - (-1 + v_c)\}^2 + \frac{1}{k+1} \{v_c - k\}^2.$$

The first term is minimized if $v_0 = v_c - 1$, the second term simplifies to $\frac{k-1}{k+1}$ and is independent of $v$, and the third term is minimized if $v_c = k$. $\qquad\square$

**Lemma B.4.** *Let $\mathcal{E}$ be an environment satisfying Assumption 3.1 with transition kernel $\{T_u\}$. Then, $\rho(\Pi_z T_\omega) < 1$ for any $z \in \mathcal{Z}$ and $\omega \in \Omega$, where $\rho$ is the spectral radius.*

*Proof.* The elements of $\Pi_z T_\omega$ are nonnegative and given by

$$(\Pi_z T_\omega)_{x',x} = \varphi_z(x')(T_\omega)_{x',x}.$$

for $x, x' \in \mathcal{X}$. The sum of values in column $x$ is thus

$$\varphi_z^\top (T_\omega)_{:,x} \leq \mathbf{1}^\top (T_\omega)_{:,x} \leq 1.$$

From this it follows that

$$\sum_{x' \neq x} (\Pi_z T_\omega)_{x',x} \leq 1 - (\Pi_z T_\omega)_{x,x}.$$

Thus, by the Gershgorin disk theorem (e.g., Axler, 2024, 5.67) we can conclude that all eigenvalues of $\Pi_z T_\omega$ must be less than or equal to 1. Now suppose, for the sake of contradiction, that 1 is an eigenvalue of $\Pi_z T_\omega$. Since we have shown that no eigenvalue can be larger, 1 is the Perron-Frobenius eigenvalue and thus there must be an eigenvector $\tilde{\nu} \in \mathbb{R}^{\mathcal{X}}$ with nonnegative components (e.g., Horn & Johnson, 2012, Theorem 8.3.1). Thus, defining $\nu \doteq \tilde{\nu}/\mathbf{1}^\top \tilde{\nu}$,

$$(\Pi_z T_\omega)\nu = \nu.$$

In other words, $\nu$ is a stationary distribution of $\Pi_z T_\omega$. However, no such distribution can exist, since we are assuming that all policies are proper (Assumption 3.1), and thus a trajectory must eventually leave $\mathcal{X}_z$. This contradiction forces us to conclude that $\rho(\Pi_z T_\omega) < 1$. $\square$

**Lemma B.5.** *Let $\mathcal{E}$ be a quasi-Markov environment. Then, $\mathcal{E}$ is rewire-robust.*

*Proof.* Let $\bar{\mathcal{E}}$ be any rewiring of $\mathcal{E}$. We will show that $\bar{\mathcal{E}} = \mathcal{E}$, which clearly implies rewire-robustness of $\mathcal{E}$. Let $(p_0, \{T_u\})$ and $(\bar{p}_0, \{\bar{T}_u\})$ be the initial state distributions and transition kernels of $\mathcal{E}$ and $\bar{\mathcal{E}}$, respectively. Our goal is to show that $\bar{p}_0 = p_0$ and $\bar{T}_u = T_u$, for all $u \in \mathcal{U}$. Let $z \in \mathcal{Z}$, and consider first the case $\Pi_z p_0 = \mathbf{0}$. We have,

$$\Pi_z p_0 = \mathbf{0} \Leftrightarrow \varphi_z^\top p_0 = 0 \Leftrightarrow \varphi_z^\top \bar{p}_0 \Leftrightarrow \Pi_z \bar{p}_0 = \mathbf{0},$$

where we have used the rewiring property (i). On the other hand, if $\Pi_z p_0 \neq \mathbf{0}$, which now implies that $\Pi_z \bar{p}_0 \neq \mathbf{0}$, rewiring property (iv) shows that we must have $\bar{\varsigma}_z = \varsigma_z = \text{span}\{\sigma_z\}$. Thus, by normalization of $\sigma_z$,

$$\Pi_z \bar{p}_0 = (\varphi_z^\top \bar{p}_0)\sigma_z = (\varphi_z^\top p_0)\sigma_z = \Pi_z p_0,$$

where we have used the rewiring property (i) and the quasi-Markov property of $\mathcal{E}$. Applying this logic to all $z \in \mathcal{Z}$ shows that $\bar{p}_0 = p_0$. We apply a similar argument to the transition kernel. Let $u \in \mathcal{U}$. Note that, to show that $\bar{T}_u = T_u$, it suffices to show that $\Pi_z \bar{T}_u \Pi_z = \Pi_z T_u \Pi_z$ and

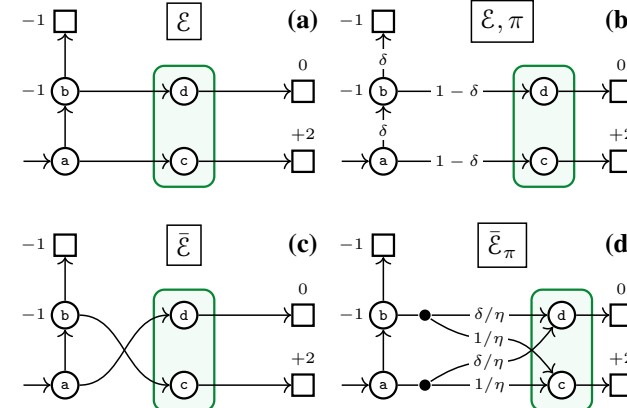

*Figure 6.* An environment that is not rewire-robust, but that is $\pi$-rewire-robust under the behavior policy $\pi$. See Lemma B.6 for details. (Here, $\eta \doteq 1 + \delta$.)

$\Pi_z \bar{T}_u \Pi_z^\perp = \Pi_z T_u \Pi_z^\perp$ for all $z \in \mathcal{Z}$. The first equality holds by rewiring property (iii). Let $x \in \mathcal{X}$. Then,

$$(\Pi_z T_u \Pi_z^\perp)_{:,x} = \mathbf{0} \iff \varphi_z^\top (T_u \Pi_z^\perp)_{:,x} = 0$$
$$\iff \varphi_z^\top (\bar{T}_u \Pi_z^\perp)_{:,x} = 0$$
$$\iff (\Pi_z \bar{T}_u \Pi_z^\perp)_{:,x} = \mathbf{0}.$$

Otherwise, by the same argument as above, we have $\bar{\varsigma}_z = \varsigma_z = \text{span}\{\sigma_z\}$, and

$$(\Pi_z \bar{T}_u \Pi_z^\perp)_{:,x} = \varphi_z^\top (\bar{T}_u \Pi_z^\perp)_{:,x} \sigma_z$$
$$= \varphi_z^\top (T_u \Pi_z^\perp)_{:,x} \sigma_z$$
$$= (\Pi_z T_u \Pi_z^\top)_{:,x},$$

where we have again used that $\sigma_z \in \Delta_{\mathcal{X}}$, that $\Phi \bar{T}_u = \Phi T_u$, and that $\mathcal{E}$ is quasi-Markov. $\square$

**Lemma B.6.** *There exists an environment $\mathcal{E}$ and a behavior policy $\pi$ satisfying Assumptions 3.1 and 3.2 such that $\mathcal{E}$ is $\pi$-rewire-robust, but not rewire-robust.*

*Proof.* Consider the environment $\mathcal{E}$ in Fig. 6a, which is a variation of Fig. 4a with modified rewards. The optimal reactive policy in $\mathcal{E}$ clearly moves right in both a and b. Figure 6c shows a rewiring $\bar{\mathcal{E}}$ of $\mathcal{E}$ in which it is optimal to move right in b, but up in a. Thus, the environment $\mathcal{E}$ is not rewire-robust. Now, consider a behavior policy $\pi$ that moves right with probability $1 - \delta$ and up with probability $1 - \delta$ (Fig. 6b). The corresponding $\pi$-rewiring is shown in Fig. 6d (cf. Fig. 4d). In the states a and b of $\bar{\mathcal{E}}_\pi$, moving right yields an expected return of $2/\eta$. In b, this is clearly preferable to moving up and terminating with a reward of $-1$. In a, the alternative is moving up and having a maximum expected return of $2/\eta - 1$. Thus, in both a and b, the optimal reactive policy moves right. Hence, $\mathcal{E}$ is $\pi$-rewire-robust. (The value of $\delta$ is irrelevant.) $\square$

**Lemma B.7.** *Let $f, g$ be two real-valued functions defined on a finite set. Then,*

$$|\max_x f(x) - \max_x g(x)| \leq \|f - g\|, \qquad (4)$$

*where $\|\cdot\|$ is the max-norm.*

*Proof.* Let $x_f$ be a maximizer of $f$. Then, $f(x_f) - g(x_f) \leq \|f - g\|$ and thus

$$\max_x f(x) = f(x_f) \leq g(x_f) + \|f - g\|$$
$$\leq \max_x g(x) + \|f - g\|.$$

The reverse direction is analogous. $\qquad\square$

## C. Proofs

In this section we present all proofs that are omitted in the main text.

**Lemma 5.1.** *Let $\mathcal{E}$ be an environment and $\pi$ a behavior policy such that Assumptions 3.1 and 3.2 hold. Then, the Markov chain $(\xi_t) \subset \Xi$ is irreducible and has a unique stationary distribution $\mu \in \Delta_\Xi$ satisfying*

$$\mu(x, \omega, x') = \mu(x, \omega)(T'_\omega)_{x',x}$$

*for all $(x, \omega, x') \in \Xi$. Furthermore, this distribution satisfies, for all $z \in \mathcal{Z}$ and $\omega \in \Omega$,*

$$\Pi_z \mu_\omega = (1 - \gamma_\pi)\pi(\omega \mid z)\Pi_z p_0 + \Pi_z T_\omega \Pi_z \mu_\omega$$
$$+ \pi(\omega \mid z) \sum_{\omega' \in \Omega} \Pi_z T_{\omega'} \Pi_z^\perp \mu_{\omega'},$$

*where $\mu_\omega \in \mathbb{R}^{\mathcal{X}}$ is defined as $\mu_\omega(x) \doteq \mu(x, \omega)$, and where $\gamma_\pi \doteq \sum_{x \in \mathcal{X}, \omega \in \Omega} \mu(x, \omega)\gamma_{x,\omega}$.*

*Proof.* We first formally define the set $\Xi$ as

$$\Xi \doteq \{(x, \omega, x') \in \tilde{\mathcal{X}} \times \Omega \times \mathcal{X}' \mid (T'_\omega)_{x',x} > 0\},$$

where

$$\tilde{\mathcal{X}} \doteq \{x \in \mathcal{X} \mid \exists \pi : \mathbb{P}_\pi\{\exists t : x_t = x \mid x_0 \sim p_0\} > 0\}$$

is the set of all reachable states. We also define $\tilde{\mathcal{Z}} \doteq \varphi(\tilde{\mathcal{X}})$ as the set of all reachable features. Irreducibility means that for any $\xi, \xi' \in \Xi$, there is a positive probability of reaching $\xi$ from $\xi'$ under the behavior policy $\pi$. Formally,

$$\mathbb{P}'_\pi\{\exists t : \xi_t = \xi \mid \xi_0 = \xi'\}$$
$$\geq \mathbb{P}'_\pi\{\exists t, t' : x'_t = x^\perp, \xi_{t+t'} = \xi \mid \xi_0 = \xi'\}$$
$$\geq \mathbb{P}_\pi\{\exists t : x'_t = x^\perp \mid \xi_0 = \xi'\} \mathbb{P}_\pi\{\exists t : \xi_t = \xi\},$$

where the second inequality follows because we have replaced the restarting kernel $\mathbb{P}'_\pi$ with the terminating kernel

$\mathbb{P}_\pi$. The first factor is 1 by Assumption 3.1. The second factor is, with $\xi \doteq (x, \omega, x')$,

$$\mathbb{P}_\pi\{\exists t : \xi_t = \xi \mid x_0 \sim p_0\}$$
$$= \mathbb{P}_\pi\{\exists t : x_t = x, \omega_t = \omega, x'_t = x' \mid x_0 \sim p_0\}$$
$$\geq \mathbb{P}_\pi\{\exists t : x_t = x \mid x_0 \sim p_0\}\pi(\omega \mid z)(T'_\omega)_{x',x}$$

which is positive by Assumption 3.2 and by the definitions of $\tilde{\mathcal{X}}$ and $\Xi$. The existence of a unique stationary distribution $\mu \in \Delta_\Xi$ follows directly from irreducibility (e.g., Rosenthal, 2006, Proposition 8.4.10). For convenience, we extend $\mu$ to the space $\Delta_{\mathcal{X} \times \Omega \times \mathcal{X}'}$ and define $\mu(x, \omega, x') = 0$ for all $(x, \omega, x') \notin \Xi$. We now show that $\mu$ can be decomposed as described. Let $\xi = (x, \omega, x') \in \Xi$. Then,

$$\mu(x, \omega)(T'_\omega)_{x',x}$$
$$= \sum_{\bar{x}' \in \mathcal{X}'} \mu(x, \omega, \bar{x}')(T'_\omega)_{x',x}$$
$$= \sum_{\bar{x}' \in \mathcal{X}'} \sum_{\xi_- \in \Xi} \mu(\xi_-)p(x, \omega \mid \xi_-)(T'_\omega)_{\bar{x}',x}(T'_\omega)_{x',x}$$
$$= \sum_{\xi_- \in \Xi} \mu(\xi_-)p(x, \omega, x' \mid \xi_-) \underbrace{\sum_{\bar{x}' \in \mathcal{X}'} p(T'_\omega)_{\bar{x}',x}}_{1}$$
$$= \mu(x, \omega, x').$$

We now show that $\Pi_z \mu_\omega$ satisfies the desired equation. Let $\xi = (x, \omega, x') \in \Xi$. Using the decomposition above and the fact that $\mu$ is stationary,

$$\mu(x, \omega) = \mu(x, \omega, x')/(T'_\omega)_{x',x}$$
$$= \sum_{\xi_- \in \Xi} \mu(x_-, \omega_-)(T'_{\omega_-})_{x'_-,x_-}p(x, \omega \mid \xi_-)$$
$$= \sum_{\substack{x_- \in \mathcal{X} \\ \omega_- \in \Omega}} \mu(x_-, \omega_-)\underbrace{\sum_{x'_- \in \mathcal{X}'} p(x, \omega \mid \xi_-)(T'_{\omega_-})_{x'_-,x_-}}_{p(x,\omega \mid x_-, \omega_-)}.$$

Thus, $\mu(x, \omega)$ is stationary with respect to $p(x, \omega \mid x_-, \omega_-)$. Plugging in the definition of the kernel $p(\xi \mid \xi_-)$, we get

$$\sum_{x'_- \in \mathcal{X}'} p(x, \omega \mid \xi_-)(T'_{\omega_-})_{x'_-,x_-}$$
$$= \sum_{x'_- \in \mathcal{X}'} p(x \mid x'_-)p(\omega \mid \xi_-, x)(T'_{\omega_-})_{x'_-,x_-}$$
$$= \sum_{x'_- \in \mathcal{X}} [z'_- = z_-][x = x_-][\omega = \omega_-](T'_{\omega_-})_{x'_-,x_-}$$
$$\quad + \sum_{x'_- \in \mathcal{X}} [z'_- \neq z_-][x = x_-]\pi(\omega \mid z)(T'_{\omega_-})_{x'_-,x_-}$$
$$\quad + p_0(x)\pi(\omega \mid z)(T'_{\omega_-})_{x^\perp,x_-}$$
$$= [z = z_-][\omega = \omega_-](T_{\omega_-})_{x,x_-}$$
$$\quad + [z \neq z_-]\pi(\omega \mid z)(T_{\omega_-})_{x,x_-}$$
$$\quad + (1 - \gamma_{x_-,\omega_-})\pi(\omega \mid z)p_0(x).$$

From this, we get

$$\mu_\omega(x) = \sum_{\substack{x_- \in \mathcal{X} \\ \omega_- \in \Omega}} \mu_{\omega_-}(x_-) p(x, \omega \mid x_-, \omega_-)$$

$$= \sum_{x_- \in \mathcal{X}} (T_\omega)_{x,x_-} [z_- = z] \mu_\omega(x_-)$$

$$+ \pi(\omega \mid z) \sum_{\substack{x_- \in \mathcal{X} \\ \omega_- \in \Omega}} (T_\omega)_{x,x_-} [z_- \neq z] \mu_{\omega_-}(x_-)$$

$$+ (1 - \gamma_\pi) \pi(\omega \mid z) p_0(x).$$

The expression for $\Pi_z \mu_\omega$ follows immediately from this by writing the sums as matrix products and the Iverson brackets as projections. $\qquad \square$

**Lemma 5.2.** *Let $\mathcal{E}$ be an environment and $\pi$ a behavior policy such that Assumptions 3.1 to 3.3 hold. Then, the iterates $Q_t$ of Algorithm 1 converge almost surely to a solution $Q_\star$ satisfying*

$$Q_\star(z, \omega) = \mathbb{E}_\mu \big[ r(z') + \max_{\omega' \in \Omega} Q_\star(z', \omega') \mid z, \omega \big]. \quad (5)$$

*Proof.* Our convergence result is based on the general stochastic approximation convergence theorem by Liu et al. (2025), specifically on a special case of their Corollary 8, which we have included as Theorem A.1. We begin by casting Algorithm 1 as a stochastic approximation procedure of the form required by Theorem A.1, and then verify that all assumptions of this theorem hold in our case. The parameter vector that is updated in Algorithm 1 is the Q-table, so in the notation of Theorem A.1, $\theta_t \doteq Q_t \in \mathbb{R}^{\mathcal{Z} \times \Omega}$. The update itself happens in Line 6 of Algorithm 1, which we can write as

$$Q_{t+1} = Q_t + \alpha_t \delta_{z_t, \omega_t} \Delta_t,$$

where $\delta_{z,\omega} \in \mathbb{R}^{\mathcal{Z} \times \Omega}$ is defined as $(\delta_{z,\omega})_{z',\omega'} \doteq [z = z'][\omega = \omega']$, and

$$\Delta_t = r(z_t') + \max_{\omega \in \Omega} Q_t(z_t', \omega) - Q_t(z_t, \omega_t).$$

If we let $\xi_t \doteq (x_t, \omega_t, x_t')$ as described above, then this recreates the required form (3) with

$$H(Q, x, \omega, x') \doteq \delta_{\varphi(x), \omega} \big\{ (r \circ \varphi)(x')$$
$$+ \max_{\omega' \in \Omega} Q\big(\varphi(x'), \omega'\big) - Q\big(\varphi(x), \omega\big) \big\}.$$

Assumption 1 of Theorem A.1 demands that $(\xi_t)$ is irreducible with stationary distribution $\mu \in \Delta_\Xi$, which is satisfied by Lemma 5.1, and Assumption 2 of Theorem A.1 is satisfied by our Assumption 3.3. We thus continue to verify Assumption 3 of Theorem A.1. Writing $z \doteq \varphi(x)$ and $z' \doteq \varphi(x')$, we define

$$H_\infty(Q, \xi) \doteq \delta_{z,\omega} \big\{ \max_{\omega' \in \Omega} Q(z', \omega') - Q(z, \omega) \big\},$$

and $b(\xi) \doteq \delta_{z,\omega} r(z')$. Let $Q \in \mathbb{R}^{\mathcal{Z} \times \Omega}$, $\xi \in \Xi$ and $c \geq 1$ be arbitrary. Then,

$$H(cQ, \xi) - cH_\infty(Q, \xi) = \delta_{z,\omega} r(z') = b(\xi),$$

as required. Moving on to Assumption 4 of Theorem A.1, we have that, for any $Q, Q' \in \mathbb{R}^{\mathcal{Z} \times \Omega}$ and $\xi \in \Xi$,

$$\|H(Q, \xi) - H(Q', \xi)\| = \|H_\infty(Q, \xi) - H_\infty(Q', \xi)\|$$
$$\leq |\max_{\omega' \in \Omega} Q(z', \omega') - \max_{\omega' \in \Omega} Q'(z', \omega')|$$
$$+ |Q(z, \omega) - Q'(z, \omega)|$$
$$\leq 2\|Q - Q'\|,$$

where $\|\cdot\|$ represents the max-norm, and where we have used Lemma B.7. We have thus shown that the Lipschitz condition in Assumption 4 of Theorem A.1 is satisfied with $L = 2$. Furthermore, both

$$h(Q) \doteq \mathbb{E}_{\xi \sim \mu} \big[ \delta_{z,\omega} \big\{ r(z') + \max_{\omega' \in \Omega} Q(z', \omega') - Q(z, \omega) \big\} \big]$$

and

$$h_\infty(Q) \doteq \mathbb{E}_{\xi \sim \mu} \big[ \delta_{z,\omega} \big\{ \max_{\omega' \in \Omega} Q(z', \omega') - Q(z, \omega) \big\} \big]$$

are finite for all $Q \in \mathbb{R}^{\mathcal{Z} \times \Omega}$. We thus continue to Assumption 5 of Theorem A.1. Let $h_c(Q) \doteq h(cQ)/c$. We need to show that $h_c \to h_\infty$ uniformly as $c \to \infty$. We have

$$\|h_c - h_\infty\| = \frac{1}{c} \big\| \mathbb{E}_{\xi \sim \mu} \big[ \delta_{z,\omega} r(z') \big] \big\| \leq \mathbb{E}_{\xi \sim \mu} \frac{|r(z')|}{c} \leq \frac{r_{\max}}{c},$$

where $r_{\max} \doteq \max_{z \in \mathcal{Z}} r(z)$. Thus, the convergence is indeed uniform, and we can consider the ordinary differential equation (ODE)

$$\dot{Q} = h_\infty(Q). \quad (6)$$

We need to show that $0 \in \mathbb{R}^{\mathcal{Z} \times \Omega}$ is a globally asymptotically stable equilibrium point of this ODE. If this the case, Theorem A.1 guarantees that that $Q_t$ converges almost surely to the globally asymptotically stable equilibrium point $Q_\star$ of

$$\dot{Q} = h(Q). \quad (7)$$

We begin with (7), since (6) is simply a special case in which the rewards are 0 everywhere. To analyze the stability of this ODE, we use a result by Borkar & Soumyanatha (1997), which we have included as Theorem A.2. For any $Q \in \mathbb{R}^{\mathcal{Z} \times \Omega}$, $z \in \mathcal{Z}$, and $\omega \in \Omega$, we define the operator $F$ as

$$(FQ)(z, \omega) \doteq \mathbb{E}_\mu \big[ r(z') + \max_{\omega' \in \Omega} Q(z', \omega') \mid z, \omega \big]$$

We can then rewrite (7) as

$$\dot{Q} = \mathbb{E}_\mu \big[ \delta_{z,\omega} (FQ - Q)(z, \omega) \big] = M \odot (FQ - Q),$$

where $M \in \mathbb{R}^{\mathcal{Z} \times \Omega}$ is defined by $M_{z,\omega} \doteq \mu(z,\omega)$ and $\odot$ denotes the Hadamard product. Thus, (7) is of the form $\dot{Q} = F_\mu Q - Q$ that is required by Theorem A.2. Here,

$$F_\mu Q \doteq Q + M \odot (FQ - Q).$$

To use Theorem A.2, we now need to verify that $F_\mu$ is max-norm nonexpansive, and that it has a unique fixed point. First note that $F$ is an episodic Bellman operator in an MDP with state space $\mathcal{Z}$, action space $\Omega$, and transition dynamics $\mu(z' \mid z, \omega)$. Thus, we know that $F$ is a weighted max-norm contraction and hence has a unique fixed point $Q_\star$ (Bertsekas, 2012, Propositions 3.3.1, 1.5.1). Furthermore, $F$ is max-norm nonexpansive, as

$$\begin{aligned}
&\|FQ - FQ'\| \\
&= \big\| \mathbb{E}_\mu \big[ \max_{\omega' \in \Omega} Q(z', \omega') - \max_{\omega' \in \Omega} Q'(z', \omega') \mid z, \omega \big] \big\| \\
&\leq \|Q - Q'\|
\end{aligned}$$

for any $Q, Q' \in \mathbb{R}^{\mathcal{Z} \times \Omega}$, which follows from Jensen's inequality and Lemma B.7. Returning to $F_\mu$, we find that this operator has the same unique fixed point as $F$:

$$F_\mu Q = Q \iff M \odot (FQ - Q) = 0 \iff Q = Q_\star,$$

where we have used that $\mu(z,\omega) > 0$ for all $z \in \tilde{\mathcal{Z}}$ and $\omega \in \Omega$. Furthermore, $F_\mu$ is max-norm nonexpansive. To see this, let $Q, Q' \in \mathbb{R}^{\mathcal{Z} \times \Omega}$, and let $\mathbf{1} \in \mathbb{R}^{\mathcal{Z} \times \Omega}$ be the all-ones matrix. Then,

$$\begin{aligned}
&\|F_\mu Q - F_\mu Q'\| \\
&= \|(\mathbf{1} - M) \odot (Q - Q') + M \odot (FQ - FQ')\| \\
&= \max_{z \in \mathcal{Z}} \max_{\omega \in \Omega} |\{1 - \mu(z,\omega)\}(Q - Q')(z,\omega) \\
&\qquad\qquad + \mu(z,\omega)(FQ - FQ')(z,\omega)| \\
&\leq \max_{z \in \mathcal{Z}} \max_{\omega \in \Omega} \{1 - \mu(z,\omega)\} \|Q - Q'\| \\
&\qquad\qquad + \mu(z,\omega) \|FQ - FQ'\| \\
&\leq \|Q - Q'\|,
\end{aligned}$$

where we have used that $\mu(z,\omega) \in (0,1)$ for all $z \in \tilde{\mathcal{Z}}$ and $\omega \in \Omega$ and that $F$ is nonexpansive. Finally, using Theorem A.2, we can conclude that the ODE (7) has a unique globally asymptotically stable equilibrium point $Q_\star$ satisfying $Q_\star = FQ_\star$. The only thing left to verify is that this fixed point is $0 \in \mathbb{R}^{\mathcal{Z} \times \Omega}$ for the ODE (6). In this case, the Bellman operator is, for $z \in \tilde{\mathcal{Z}}$ and $\omega \in \Omega$,

$$(F_\infty Q)(z,\omega) \doteq \mathbb{E}_\mu \big[ \max_{\omega' \in \Omega} Q(z', \omega') \mid z, \omega \big].$$

It is easily seen that $F_\infty 0 = 0$. As we have already shown that the fixed point is unique, we are done. Theorem A.1 now guarantees that the iterates $Q_t$ of Algorithm 1 converge almost surely to $Q_\star$ which satisfies (2). $\qquad\square$

**Lemma 5.4.** *Let $\mathcal{E}$ be an environment and $\pi$ a behavior policy such that Assumptions 3.1 and 3.2 hold. Then, if $\hat{\mathcal{M}}_\pi$ is the corresponding $\pi$-MDP with transition kernel $\{\hat{T}_\omega\}$,*

$$\mu(z' \mid z, \omega) = (\hat{T}_\omega)_{z', z}$$

*for all $z, z' \in \mathcal{Z}$, and all $\omega \in \Omega$.*

*Proof.* Let $z, z' \in \mathcal{Z}$ and $\omega \in \Omega$. Then, by the definition of $\mu(z, \omega, z')$,

$$\begin{aligned}
\mu(z' \mid z, \omega) &= \frac{\sum_{x \in \mathcal{X}_z, x' \in \mathcal{X}_{z'}} \mu(x, \omega, x')}{\sum_{x \in \mathcal{X}_z} \mu(x, \omega)} \\
&= \frac{\sum_{x' \in \mathcal{X}} \varphi_{z'}(x') \sum_{x \in \mathcal{X}_z} (T_\omega)_{x', x} \mu_\omega(x)}{\sum_{x \in \mathcal{X}} \varphi_z(x) \mu_\omega(x)} \\
&= \varphi_{z'}^\top \bar{T}_\omega \frac{\Pi_z \mu_\omega}{\varphi_z^\top \mu_\omega},
\end{aligned}$$

where we have used the rewiring property of $\bar{\mathcal{E}}_\pi$ in the last step. The transition kernel of $\hat{\mathcal{M}}_\pi$ is $(\hat{T}_\omega)_{z',z} = \varphi_{z'}^\top \bar{T}_\omega \psi_z^\omega$. Thus, we need to prove that $\psi_z^\omega = \Pi_z \mu_\omega / \varphi_z^\top \mu_\omega$ for all $z \in \mathcal{Z}$ and $\omega \in \Omega$. As $\psi_z^\omega$ is normalized, it is enough to prove that $\psi_z^\omega \propto \Pi_z \mu_\omega$. Using the fact that $\Pi_z T_\omega \Pi_z = \Pi_z \bar{T}_\omega \Pi_z$, we have, from the definition of $\tilde{\sigma}_z$ and Lemma 5.1,

$$\pi(\omega \mid z) \tilde{\sigma}_z = (I - \Pi_z \bar{T}_\omega) \Pi_z \mu_\omega.$$

The final result follows as

$$\psi_z^\omega \propto (I - \Pi_z \bar{T}_\omega)^{-1} \tilde{\sigma}_z \propto \Pi_z \mu_\omega. \qquad\square$$

## D. The Bellman risk is learnable

Sutton & Barto (2018, Section 11.6) show that the value error and Bellman error are not *learnable*, meaning that they cannot generally be estimated purely from observed quantities such as features and rewards. The proof constructs partially observable environments which cannot possibly be distinguished based on the observed features and rewards, but in which the value errors and Bellman errors do not coincide. While the value error at least has a unique minimizer, Sutton & Barto show that the minimizer of the Bellman error can depend on these unobservable differences between environments. In this section, we show that this shortcoming does not apply to the value risk $\mathcal{R}_V$ or Bellman risk $\mathcal{R}_B$ defined in Section 4. The fundamental reason is that both $\mathcal{R}_V$ and $\mathcal{R}_B$ are defined purely based on observable quantities (rewards and features). In the following, we remove the assumption that $\varphi$ is deterministic. We thus consider the general POMDP setting with stochastic observations $z_t \sim \varphi(\cdot \mid x_t)$. Since these results are independent of the behavior policy, we simply let the environments be *autonomous*, meaning that there are fixed (latent) transition dynamics $p(x_{t+1} \mid x_t)$. Following Sutton & Barto, we say that the *data distributions* of two autonomous environments coincide, if the distributions over feature sequences

$p(z_0, z_1, \dots)$ are the same. We assume that the reward is a deterministic function of the feature; this is without loss of generality, since the feature space could simply be extended to include the observed reward. In the following result, we assume that the state distributions of the autonomous environments converge to their stationary distributions.

**Proposition D.1.** *Let $\{\mathcal{E}_i\}$ be a set of autonomous environments with stationary distributions $\{\mu_i \in \Delta_{\mathcal{X}_i}\}$ satisfying $\mathbb{P}_i\{x_t = x\} \to \mu_i(x)$ as $t \to \infty$. If the data distributions of $\{\mathcal{E}_i\}$ coincide, then $\mathcal{R}_B^i(v) = \mathcal{R}_B^j(v)$ and $\mathcal{R}_V^i(v) = \mathcal{R}_V^j(v)$, for any two environments $i, j$ and any $v \in \mathbb{R}^{\mathcal{Z}}$.*

*Proof.* The value risk and Bellman risk are both defined in terms of expected values of the form $\mathbb{E}_\mu[f(z_t)]$, where $\mu$ is the stationary distribution over the state:

$$\mathbb{E}_{\mu_i}[f(z_t)] = \sum_{x \in \mathcal{X}_i} \mu_i(x) \sum_{z \in \mathcal{Z}} \varphi_i(z \mid x) f(z),$$

where we see that the expression is independent of $t$. We prove the result by showing that $\mathbb{E}_{\mu_i}[f(z_t)] = \mathbb{E}_{\mu_j}[f(z_t)]$ for all $i, j$, and $f$. Consider the following related quantity:

$$\begin{aligned}
\mathbb{E}_i[f(z_t)] &= \sum_{z \in \mathcal{Z}} \mathbb{P}_i\{z_t = z\} f(z) \\
&= \sum_{x \in \mathcal{X}} \mathbb{P}_i\{x_t = x\} \sum_{z \in \mathcal{Z}} \varphi_i(z \mid x) f(z).
\end{aligned}$$

As the data distributions of $\{\mathcal{E}_i\}$ coincide, we clearly have $\mathbb{E}_i[f(z_t)] = \mathbb{E}_j[f(z_t)]$ for any $t$. Thus, this equality must also hold in the limit as $t \to \infty$. From our assumption that $\mathbb{P}_i\{x_t = x\} \to \mu_i(x)$ as $t \to \infty$, we have

$$\begin{aligned}
\mathbb{E}_{\mu_i}[f(z_t)] &= \lim_{t \to \infty} \mathbb{E}_i[f(z_t)] \\
&= \lim_{t \to \infty} \mathbb{E}_j[f(z_t)] = \mathbb{E}_{\mu_j}[f(z_t)],
\end{aligned}$$

which concludes the proof. $\square$

Note that the above conclusion does not extend to the Value error $\overline{\text{VE}}$ and Bellman error $\overline{\text{BE}}$, as these are defined as expected values of the form $\mathbb{E}_\mu[f(x_t)]$. The proof above also establishes that the value risk $\mathcal{R}_V$ and the Bellman risk $\mathcal{R}_B$ are "learnable" if the state distribution converges to the stationary distribution. An asymptotically unbiased estimate can be obtained by a simple estimator of the form

$$\hat{\mathcal{R}}_t \doteq \frac{1}{t} \sum_{\tau=1}^t f(z_\tau),$$

where $f$ is chosen as above to reflect either $\mathcal{R}_V$ or $\mathcal{R}_B$. This estimator is furthermore simple to implement in a recursive fashion. The following result shows that the value error and the value risk are closely related.

**Proposition D.2.** *The value risk has the same minimizer as the value error.*

*Proof.* We first introduce the *return error*,

$$\mathcal{R}(v) = \mathbb{E}_\mu\big[\{v(z_t) - R_t\}^2\big].$$

It is a well-known property of mean squared risk measures like this that the minimizer of the risk is the *regression function*, in this case $v(z) = \mathbb{E}_\mu[R_t \mid z_t = z]$. Looking at the definition of the value risk, it is obvious that this quantity also minimizes the value risk. That the value error and the return error are minimized by the same function has already been shown by Sutton & Barto (2018). $\square$

# E. On $q_\star$-realizability

We first show that our definition of $q_\star$-realizability (Definition 3.4) is the appropriate definition of linear $q_\star$-realizability for the case of hard state aggregation and stochastic shortest path problems that we consider. Note that in this case, the optimal action-value function is stationary: $q_{h_1}^\star = q_{h_2}^\star$, for any $h_1$ and $h_2$. Furthermore, the feature mapping $\varphi$ is a hard state aggregation. We can write this in the notation of Weisz et al. (2021) by defining the mapping $\tilde{\varphi} : \mathcal{X} \times \mathcal{U} \to \mathbb{R}^d$, where $d = |\mathcal{Z} \times \mathcal{U}|$, and $\tilde{\varphi}(x, u)$ is the one-hot vector in $\mathbb{R}^d$ corresponding to the tuple $(\varphi(x), u)$. Then, the function $q : \mathcal{Z} \times \mathcal{U} \to \mathbb{R}$ in Definition 3.4 is equivalent to the vector $\theta^\star$ defined in Assumption 1 of Weisz et al. (2021), since $\langle \tilde{\varphi}(x, u), \theta^\star \rangle = (\theta^\star)_{\varphi(x), u} = q(\varphi(x), u)$.

**Proposition E.1.** *Let $\mathcal{E}$ be a $q_\star$-realizable environment. Then, $\mathcal{E}$ is generalized rewire-robust.*

*Proof.* Let $q_\star : \mathcal{X} \times \mathcal{U} \to \mathbb{R}$ be the optimal action-value function in $\mathcal{M}$, the MDP underlying $\mathcal{E}$. As $\mathcal{E}$ is $q_\star$-realizable, there exists a function $q : \mathcal{Z} \times \mathcal{U} \to \mathbb{R}$ such that

$$q_\star(x, u) = q\big(\varphi(x), u\big)$$

for all $x \in \mathcal{X}$ and $u \in \mathcal{U}$. Let $\bar{\mathcal{E}}$ be any generalized rewiring of $\mathcal{E}$, and let $x \in \mathcal{X}$ and $u \in \mathcal{U}$ be arbitrary. Then, by Bellman optimality of $q$ in $\mathcal{M}$,

$$\begin{aligned}
&q(\varphi(x), u) \\
&= \sum_{x' \in \mathcal{X}} (T_u)_{x',x}\big\{(r \circ \varphi)(x') + \max_{u' \in \mathcal{U}} q\big(\varphi(x'), u'\big)\big\} \\
&= \sum_{z' \in \mathcal{Z}} (\varphi_{z'}^\top T_u)(x)\big\{r(z') + \max_{u' \in \mathcal{U}} q(z', u')\big\} \\
&= \sum_{z' \in \mathcal{Z}} (\varphi_{z'}^\top \bar{T}_u)(x)\big\{r(z') + \max_{u' \in \mathcal{U}} q(z', u')\big\} \\
&= \sum_{x' \in \mathcal{X}} (\bar{T}_u)_{x',x}\big\{(r \circ \varphi)(x') + \max_{u' \in \mathcal{U}} q\big(\varphi(x'), u'\big)\big\},
\end{aligned}$$

where we have used the rewiring property $\Phi T_u = \Phi \bar{T}_u$. Thus, $q$ also represents the optimal action-value function in $\bar{\mathcal{M}}$. It follows that an optimal policy in $\bar{\mathcal{E}}$ is greedy with respect to $q$, just like in $\mathcal{E}$. Thus, $\mathcal{E}$ is generalized rewire-robust. $\square$

