# OpenReview forum: "Commit to the Bit: Reactive Reinforcement Learning Done Right"
_ICML.cc/2026/Conference — ICML 2026 regular_

### Official Review · Reviewer_RLJq · 2026-02-25

**Soundness:** 3
**Presentation:** 3
**Significance:** 2
**Originality:** 3
**Overall Recommendation:** 5
**Confidence:** 2

**Summary:**

This paper introduces Committed Q-Learning, a novel algorithm with asymptotic convergence guarantees to the best reactive policy in non-Markovian environments satisfying a new structural assumption dubbed robust rewiring, which is weaker than conditions in previous works, such as $q^\star$ realizability.

**Compliance With Llm Reviewing Policy:**

Affirmed.

**Final Justification:**

I am happy to keep supporting the acceptance of the paper.

**Key Questions For Authors:**

1) Can you please distinguish your notion of $q^\star$ realizability with the one used in other RL theory works such as [1]. I think that it could be very beneficial to add a discussion about this in the main text to avoid any risk of confusion between the two terminology.

2) Out of curiosity, would it be possible to prove finite-time convergence guarantees for Committed Q learning if augmented with exploration bonuses? Following in spirit the analysis in [2].

[1] Exponential Lower Bounds for Planning in MDPs With Linearly-Realizable Optimal Action-Value Functions. Weisz, Amortila and Szepesvari.

[2] Is Q-learning Provably Efficient? Chi Jin, Zeyuan Allen-Zhu, Sebastien Bubeck, Michael I. Jordan

**Limitations:**

The only limitation I would mention is that the convergence results are only asymptotics and do not come with a rate.
The result is still very interesting.

**Strengths And Weaknesses:**

The paper is well written and clear. I appreciated in particular that one could understand the proof technique from the main text as many proofs are reported in their completed form.

---

> ### Author Rebuttal · Authors · 2026-03-31
>
> Thank you very much for taking the time to review our paper, and for your positive evaluation! Below, we respond to your critical points and questions.
>
> > 1. Can you please distinguish your notion of $q_\star$-realizability with the one used in other RL theory works such as [1]. I think that it could be very beneficial to add a discussion about this in the main text to avoid any risk of confusion between the two terminology.
>
> Our definition is equivalent to the definition in [1].
> We consider stochastic shortest path problems, so the value function is stationary ($q_{h_1}^\star = q_{h_2}^\star$ for any $h_1$ and $h_2$).
> Furthermore, the feature mapping $\varphi$ is a hard state aggregation.
> We can write this in the notation of [1] by defining the mapping $\tilde{\varphi}: \mathcal X \times \mathcal U \to \mathbb R^d$, where $d = |\mathcal Z \times \mathcal U|$, and $\tilde\varphi(x, u)$ is the one-hot vector in $\mathbb R^d$ corresponding to the tuple $(\varphi(x), u)$.
> Then, the function $q: \mathcal Z \times \mathcal U \to \mathbb R$ defined in Definition 3.7 of our paper is equivalent to the vector $\theta^\star$ defined in Assumption 1 of [1], since $\langle\tilde\varphi(x, u), \theta^\star\rangle = (\theta^\star)_{\varphi(x), u} = q(\varphi(x), u)$.
> We are happy to include this explanation in the paper, if you think that it clarifies the notation.
>
> > 2. Out of curiosity, would it be possible to prove finite-time convergence guarantees for Committed Q learning if augmented with exploration bonuses? Following in spirit the analysis in [2].
>
> This is a very interesting question that we have also considered.
> We don't see why an analysis along the lines of [2] should not be applicable to Committed Q-learning, but leave this direction for future work, as the additional complexities that this setting brings with it makes this go beyond the scope of our paper.
>
> > The only limitation I would mention is that the convergence results are only asymptotics and do not come with a rate. The result is still very interesting.
>
> This is a valid point that applies to most theoretical convergence results on Q-learning.
> In our numerical experiments in the corridor environment (see [here](https://anonymous.4open.science/r/icml2026-rebuttal-q-commit/README.md)), which we will include in the final version of this paper, we found that $(Q_t)$ converges asymptotically as $\mathcal O(1/t)$.
> We agree that a theoretical analysis of the convergence rate would be very interesting, but leave this for future work.

---

> > ### Author Rebuttal · Reviewer_RLJq · 2026-04-01
> >
> > Dear authors,
> >
> > Thank you for your rebuttal!
> >
> > 1. Thank you for the clarifications. Please include in the final version.
> >
> > 2. After the clarification 1., I think that an exploratory version of Committed Q Learning with exploration bonuses will require linearity of the dynamics and rewards. linearity of $q^\star$ should not be enough.
> >
> > 3. Thank you for the additional experiment.
> >
> > I keep my very positive score.
> >
> > Best,
> > reviewer

---

### Official Review · Reviewer_UrQx · 2026-03-08

**Soundness:** 4
**Presentation:** 3
**Significance:** 3
**Originality:** 4
**Overall Recommendation:** 5
**Confidence:** 4

**Summary:**

The paper studies a reinforcement learning (RL) problem on partially observable Markov decision processes (POMDPs) where the observations are deterministic, i.e., multiple states map to the same observation, and the objective is to aim to learn a memoryless policy, which is a function of the current observation. The authors develop an algorithm called committed Q-learning, where the policy “commits” to an action and does not resample it until the observed feature changes. The authors provide that this algorithm converges to an optimal memoryless policy under the “rewire-robust” condition, where, based on the prior state of entering an observation, the policy is still optimal.

**Compliance With Llm Reviewing Policy:**

Affirmed.

**Final Justification:**

I am still in favor of accepting the paper based on the author's rebuttal to my and other reviewer's reviews.

**Key Questions For Authors:**

1- Does this setting hold on a memory-based policy? I am aware that you can expand the POMDP with finite-state controllers, but I am not sure whether the rewire-robust condition holds with finite-state controllers that use memory.

2- How does this relate to deep RL with partial observability in practice? Is it possible to use the commitment approach on certain parts of the state or observations based on theory instead of adding more features or memory states to the RL policy?

3- On a similar question to above, is it possible to extend this idea, even with simple function approximation techniques that do not rely on neural networks?

4- Also, I would like to have a comparison, even though it may be conceptual,  between different (standard) POMDP methods, such as point-based value iterations, methods based on finite-state controllers or policy-gradient based methods with shallow or deep neural-network-based policies.

**Limitations:**

yes

**Strengths And Weaknesses:**

Strengths:

The paper deals with a tabular setting, and also the introduced rewire-robust condition also makes sense and the authors explain it in detail. I am aware that computing or learning optimal policies in POMDPs is intractable, so I appreciate when a work introduces a set of conditions or restrictions such that the resulting setting is tractable.

The algorithm is also very simple, where a new action is only sampled if the observation is “no-committed”, which makes the analysis easier.

The authors also use the Bellman risk instead of value functions, and demonstrate that this risk can be learned in this setting, and I am aware that learning value functions in POMDPs is generally intractable, so I appreciate any such insight for POMDPs.

Weakness:

As with many similar papers or settings, the authors had to restrict the problem setup with tabular states and actions, and also deterministic observations, which I acknowledge that it is a necessary condition for proving convergence results.

The authors can also discuss how the limited exploration on the “commitment” phase can lead to slower convergence in practice, even though it’s necessary for proving the results.

Finally, I think the authors can demonstrate the algorithm, even on a tabular setting, showing the comparions between the classical Q-learning, or the proposed algorithm with different step sizes.

---

> ### Author Rebuttal · Authors · 2026-03-31
>
> Thank you very much for taking the time to review our paper, and for your positive evaluation! Below, we respond to your critical points and questions.
>
> > 1- Does this setting hold on a memory-based policy? I am aware that you can expand the POMDP with finite-state controllers, but I am not sure whether the rewire-robust condition holds with finite-state controllers that use memory.
>
> This is an important question that is closely related to our original motivation for working on this problem: given a finite memory mechanism, how can we learn an optimal policy?
> A finite state controller is a combination of a finite memory and a reactive policy.
> We give an example of this setup in Figure 2, where we show how to apply this idea to the partially observable T-maze environment.
> In general, a finite memory $\varphi$ describes a mapping from histories to features, $\varphi: \mathcal Y^* \to \mathcal Z$.
> This is a state aggregation of the history-MDP with state space $\mathcal Y^\*$.
> The concept of rewire-robustness can be applied to this setting: if the optimal action for each history corresponding to the same memory state $z$ is identical, then the POMDP with this memory mechanism is rewire-robust.
> The only complication that arises here is that the state space $\mathcal Y^*$ of the history MDP is generally not finite.
> In some cases, such as the T-maze, this is not a problem, since the memory we show in Figure 2 also defines a state aggregation on the finite state space $\mathcal X$.
> We believe that the conclusions of our results likely carry over to the more general setting of countable state spaces, but the additional complexities that this setting brings with it makes this go beyond the scope of our paper.
>
> > 2- How does this relate to deep RL with partial observability in practice? [...]
>
> This is an interesting question that we do not know the answer to yet.
> We are currently exploring possible applications of our theory to deep reinforcement learning algorithms.
>
> > 3- On a similar question to above, is it possible to extend this idea, even with simple function approximation techniques that do not rely on neural networks?
>
> The most basic extension would be to soft state aggregation (or, equivalently, stochastic observations).
> This is the setting we originally hoped to address.
> Unfortunately, it does not seem as though this extension is very straightforward.
> Our theory is fundamentally based on the idea of features as "sets of states," which really only applies to hard state aggregation.
> Similarly, the idea of "commitment" relies on the concept of "entering" and "exiting" a feature, the meaning of which is less clear beyond hard state aggregation.
> We still think that these are important questions to pursue, and are currently trying to extend our theory into this direction.
>
> > 4- Also, I would like to have a comparison, even though it may be conceptual, between different (standard) POMDP methods, such as point-based value iterations, methods based on finite-state controllers or policy-gradient based methods with shallow or deep neural-network-based policies.
>
> With the exception of point-based value iteration, which is an exact planning algorithm for POMDPs, these methods are not very well understood theoretically, and are thus difficult to compare to our results.
> Furthermore, these methods try to address the complete POMDP learning problem, while we only address the problem of learning a reactive policy, and leave the complementary problem of learning a memory mechanism for future work.
>
> > The authors can also discuss how the limited exploration on the “commitment” phase can lead to slower convergence in practice, even though it’s necessary for proving the results.
>
> This is an interesting point, as we actually found the opposite effect.
> The goal of the Q-learning algorithm is to find a deterministic policy (mapping features to options).
> Any deterministic policy is automatically "committed."
> Thus, only exploring committed policies during training actually speeds up convergence.
> As a concrete example of this, consider the corridor environment introduced in Figure 1.
> When entering the corridor, committed Q-learning chooses action "left" or "right," to be played until the corridor is exited.
> Thus, the corridor is exited either immediately, or after $k$ steps, where $k$ is the corridor length.
> In contrast, a uniform non-committed policy that selects left or right at every step will only reach the end of the corridor after about $k^2$ steps on average.
>
> > Finally, I think the authors can demonstrate the algorithm, even on a tabular setting, showing the comparions between the classical Q-learning, or the proposed algorithm with different step sizes.
>
> To verify our theoretical findings empirically, we ran small-scale experiments in the corridor environment discussed in the paper.
> The results can be found [here](https://anonymous.4open.science/r/icml2026-rebuttal-q-commit/README.md).

---

> > ### Author Rebuttal · Reviewer_UrQx · 2026-04-02
> >
> > Thanks for the detailed response, I am happy to keep my accept score.

---

### Official Review · Reviewer_VKqe · 2026-03-12

**Soundness:** 3
**Presentation:** 2
**Significance:** 2
**Originality:** 2
**Overall Recommendation:** 4
**Confidence:** 3

**Summary:**

This paper introduces a new Q-learning algorithm, Committed Q-learning, for finite environments with deterministic observations. The key idea of the proposed algorithm is that the agent samples one option and commits to it until the observed feature changes. The authors also provide several analyses and proofs of theorems (rewire-robustness and convergence of Committed Q learning). The proposed algorithm is simple and easy to understand.

**Compliance With Llm Reviewing Policy:**

Affirmed.

**Final Justification:**

The authors address all of my questions.

**Key Questions For Authors:**

The questions discussed under the Weaknesses section capture the main concerns.

**Limitations:**

The contribution is limited due to the deterministic action setting.

**Strengths And Weaknesses:**

**Strengths**
1. The authors address an important issue related to Q-learning and optimization problems in the RL community.
2. The framework connecting the observed process to an aggregate MDP via entrance distributions is well-justified.


**Weaknesses**
1. The theory relies on rewire-robustness (Definition 3.5) to transfer optimality from the rewired quasi-Markov environment back to the original environment (Theorem 3.6, Lemma 5.3, Corollary 5.6). Although this condition is weaker than q$^\*$-realizability, it remains difficult to verify or guarantee in practice, since practitioners would need to reason about whether the relevant rewiring preserves the optimal reactive policy.
2. The algorithm requires a fixed, finite option set. Optimizing over options may be nontrivial, and the paper provides little guidance on option design or required size. Although Figure 2 provides an example, could the authors include toy experimental examples (compared to Q-learning), such as MiniGrid [1]?
3. Although the authors leave stochastic options for future research, the current contribution seems limited because the Committed Q-learning only considers deterministic options. The examples are also tailored to deterministic settings, and extending the framework to stochastic options would likely require stronger assumptions. Thus, the novelty appears somewhat limited in its current form.


[1] Chevalier-Boisvert, Maxime, et al. "Minigrid & miniworld: Modular & customizable reinforcement learning environments for goal-oriented tasks." Advances in Neural Information Processing Systems 36 (2023): 73383-73394.

---

> ### Author Rebuttal · Authors · 2026-03-31
>
> Thank you very much for taking the time to review our paper. Below, we respond to your critical points and questions.
>
> **Rewire-robustness.**
> Our general convergence result (Corollary 5.6) shows that committed Q-learning in an environment $\mathcal E$ with behavior policy $\pi$ converges to a policy that is optimal in $\bar{\mathcal E}\_\pi$.
> We agree that it may be difficult in practice to verify whether an optimal policy in $\bar{\mathcal E}\_\pi$ is optimal in $\mathcal E$.
> The introduction of *rewire-robustness* (Definition 3.5) makes this verification easier.
> Intuitively, an environment is rewire-robust, if the optimal action in each state $x$ only depends on the feature $\varphi(x)$.
> This is both less restrictive than $q_\star$-realizability (Proposition 3.8), and is much easier to verify, since a practitioner does not need to reason about value functions!
>
> Littman (1994) showed that finding a reactive policy in the setting that we consider is NP-hard in general, meaning that some kind of assumption is required for efficient algorithms.
> To the best of our knowledge, rewire-robustness is the weakest such assumption introduced to date.
> Previous work has shown that value iteration and Q-learning also work if the Markov property does not hold, i.e., if the state is partially observable, as long as the environment is $q_\star$-realizable.
> This property, which states that the optimal Q-values of all states inside a given feature are identical, is very natural.
> If the states within features have different values, then which value should be assigned to a feature?
> How can dynamic programming possibly work if we can't even define the value function clearly?
> Realizability circumvents this issue since, in this case, there is only one sensible value for each feature.
> In this paper, we establish the important result that $q_\star$-realizability is not necessary: dynamic programming works in a much more general non-Markovian setting, in which there is no clear candidate value function!
> We show that in _any_ partially observable (tabular) environment, committed Q-learning converges to feature values that represent a mixture of the "entrance state" values.
> Our key insight (Lemma 4.3) shows that in rewire-robust or quasi-Markov environments, a greedy policy with respect to these values is optimal.
> We believe that this represents a significant and highly original contribution.
>
> **Empirical results.**
> To verify our theoretical findings empirically, we ran small-scale experiments in the corridor environment discussed in the paper.
> The results can be found [here](https://anonymous.4open.science/r/icml2026-rebuttal-q-commit/README.md).
> An extension to more complex environment like MiniGrid goes beyond the scope of this paper, since these environments are designed as test beds for deep reinforcement learning algorithms with memory.
> Treating them as tabular, especially when adding memory (e.g., via frame-stacking), leads to a prohibitively large feature space.
>
> **Stochastic policies.**
> In partially observable environments, deterministic policies can be arbitrarily worse than stochastic policies, as was first shown by Singh et al. (1994b).
> For this reason, we explicitly phrase all our results in terms of stochastic distributions over actions (which we call options).
> We are thus somewhat confused over the statements
>
> > Although the authors leave stochastic options for future research, the current contribution seems limited because the Committed Q-learning only considers deterministic options. The examples are also tailored to deterministic settings, and extending the framework to stochastic options would likely require stronger assumptions. Thus, the novelty appears somewhat limited in its current form.
>
> and
>
> > The contribution is limited due to the deterministic action setting.
>
> One limitation of our work, which we also discuss in the conclusion of our paper, is that we require a finite set of options (action distributions) to make the "maximization" step of the algorithm (in the Q-table update) feasible.
> However, without an additional assumption like this, the maximization is generally NP-hard, as was shown by Vlassis et al. (2012).
>
>
> We hope that we could address your concerns and answer your questions, and would be grateful about an increased score unless you have further concerns.

---

> > ### Author Rebuttal · Reviewer_VKqe · 2026-04-03
> >
> > Thank you to the authors for carefully addressing all of my questions and for clarifying several points that I had previously misunderstood. I appreciate the additional experimental discussion and encourage the authors to incorporate these results clearly in the revised version.
> >
> > I am happy to raise my score from negative to positive.

---

### Official Review · Reviewer_yqeK · 2026-03-15

**Soundness:** 4
**Presentation:** 4
**Significance:** 3
**Originality:** 2
**Overall Recommendation:** 4
**Confidence:** 5

**Summary:**

This is a theory paper.  It proposes a new algorithm called committed Q-learning for partially observable domains where actions are selected based on the last observation only.  This algorithm ensures that the agent sticks with the same action (or option) whenever the agent receives the same observation in subsequent time steps.  The paper formalizes a notion of quasi-Markov decision processes and derives a theory that demonstrates that committed Q-learning converges to the optimal policy under certain assumptions.

**Compliance With Llm Reviewing Policy:**

Affirmed.

**Final Justification:**

I now understand that the action is selected by a behavioural policy that is stochastic. Furthermore, the algorithm converges to the optimal Q-function as opposed to the Q-function of the behavioural policy. Hence, exploration can be ensured by the behavioural policy without impacting the convergence proof. The authors also demonstrated that committed Q-learning can find an optimal policy in some environment where regular Q-learning completely fails. My concerns are addressed and I am raising my score to weak accept.

**Key Questions For Authors:**

1.  How would you extract a stochastic policy from the Q-values in committed Q-learning?
2.  What exploration strategy would you use in committee Q-learning?
3.  How would you guarantee convergence for this exploration strategy?
4.  How does your theoretical analysis take into account exploration?
5.  Can you demonstrate empirically that committed Q-learning yields different results that plain Q-learning in some important class of problems?

**Limitations:**

The paper does not discuss any limitation.  I encourage the authors to acknowledge that committed Q-learning may not yield different results than plain Q-learning in practice.

**Strengths And Weaknesses:**

Strengths:
* This is a theoretically strong paper.  It provides an in-depth theoretical analysis of the effects of committing to an action when the agent receives the same observation in subsequent time steps.
* Presentation: The paper is clear and well written

Post-rebuttal:  the weaknesses described below have been addressed by the rebuttal and therefore are not a concern anymore.

Weaknesses:
* Significance and originality: limited.  Committed Q-learning differs from Q-learning in only one line, Line 9, which says that the agent should stick with the same action when the observation does not change.  initially, I did not think that this is a difference because a policy that maps observations to actions would always select the same action when receiving the same observation.  After thinking about this more, I realized that there is indeed a difference when the policy is stochastic.  In that case, the agent would sample an action from the same distribution when receiving the same observation, but the sampled action may be different.  This matters only when the policy is stochastic.  Unfortunately, the pseudocode for committed Q-learning is incomplete as it does not indicate how a policy is extracted from the updated Q-values.  A naive approach would be to simply define $\pi(z) = argmax_u Q(z,u)$, which yields a deterministic policy and then there is no difference between committed Q-learning and regular Q-learning.  However such an approach would not explore and therefore may not converge to the optimal policy.  Since most exploration techniques (e.g., epsilon-greedy, Boltzmann exploration) introduce some noise the policy becomes stochastic and Line 9 does indeed yield a difference.  However, the paper does not discuss any exploration strategy.  Furthermore, exploration strategies that guarantee convergence have the property that they become greedy in the limit and therefore will determinize the action choice in the limit (unless there are several optimal actions for the same observation).  Assumption 3.2 indicates that the policy must be stochastic to ensure exploration, however this is insufficient to ensure convergence.  For convergence, the probability of exploration needs to vanish in the limit.    My main concern with the paper is that it studies a very narrow question that impacts exploration and convergence without studying exploration and convergence.  Despite the fact that the paper is entirely theoretical, the theoretical analysis is incomplete because it omits any discussion of exploration and the consequences for convergence.
* Lack of empirical analysis:  I realize that this is a theory paper and I am not against the publication of theory only.  However, in this particular case, it is not clear that the tiny difference (Line 9) studied in the paper makes a difference in practice.  In the absence of this paper, the community would simply use regular Q-learning with an exploration strategy that determinizes the choice of action in the limit for convergence.  I personally feel that this will likely yield the same results as committed Q-learning in most problems.  However, if this is not the case, then this should be demonstrated empirically by the paper to justify this in-depth analysis of a narrow issue.  Otherwise, the community will simply ignore the work.

Soundness:  I did not verify all the theoretical derivations, but they look right to me. Ultimately, I did not feel compelled to verify the derivations because the paper does not make a strong case for the study of such a narrow issue.

---

> ### Author Rebuttal · Authors · 2026-03-31
>
> Thank you very much for taking the time to review our paper. Below, we respond to your critical points and questions.
>
> **Exploration.**
> This seems to be your main concern: how are actions chosen in the Committed Q-learning algorithm, and how do we guarantee convergence if the policy needs to be exploratory?
> You write:
> > the pseudocode for committed Q-learning is incomplete as it does not indicate how a policy is extracted from the updated Q-values.
>
> As written in the first line of the pseudocode (Algorithm 1), the behavior policy $\pi$ is given as an input to the algorithm.
> If the algorithm were to update this policy, then your following comments, discussing techniques such as epsilon-greedy exploration, would indeed be very important to discuss in the paper.
> However, $\pi$ is not updated by the algorithm, and exploration is guaranteed by Assumption 3.2.
> Our convergence result (Theorem 3.6) characterizes the convergence of the Q-function $Q_t$ to a solution $Q_\star$ and the optimality of the greedy policy with respect to this solution.
> This greedy policy is different from the behavior policy $\pi$ used to select actions in the algorithm.
> In our new experiments (see below), we additionally evaluate the algorithm with an epsilon-greedy policy.
>
> In the theoretical literature on Q-learning, this type of analysis is standard: fix an exploration strategy (such as the fixed behavior policy $\pi$), and only analyze the convergence of the Q-function.
> While we believe that the conclusion of our theorem would likely extend to an adaptive setting (such as the epsilon-greedy GLIE strategy that you describe), a rigorous proof of this would require an augmented analysis in which the Markov chain $(\xi_t)$ described in section 5 is non-stationary due to the evolving policy.
> We agree that such an analysis would be very interesting, but the additional complexities that this setting brings with it makes this go beyond the scope of our paper.
>
> **Empirical results.**
> > it is not clear that the tiny difference (Line 9) studied in the paper makes a difference in practice
>
> To verify our theoretical findings empirically, we ran small-scale experiments in the corridor environment discussed in the paper.
> The results can be found [here](https://anonymous.4open.science/r/icml2026-rebuttal-q-commit/README.md).
> In the corridor environment, committed Q-learning converges to the optimal policy, while non-committed Q-learning does not.
>
> **Significance & originality.**
> We would like to stress that this paper is not meant to improve the state of the art in empirical reinforcement learning.
> Instead, our goal is to better understand the foundations that these algorithms are built on.
> Q-learning is a stochastic approximation of the value iteration algorithm, which is fundamentally based on the Markov property.
> Previous work has shown that value iteration and Q-learning also work if the Markov property does not hold, i.e., if the state is partially observable, as long as the environment is *$q_\star$-realizable*.
> This property, which states that the optimal Q-values of all states inside a given feature are identical, is very natural.
> If the states within features have different values, then which value should be assigned to a feature?
> How can dynamic programming possibly work if we can't even define the value function clearly?
> Realizability circumvents this issue since, in this case, there is only one sensible value for each feature.
> In this paper, we establish the important result that $q_\star$-realizability is not necessary: dynamic programming works in a much more general non-Markovian setting, in which there is no clear candidate value function!
> We show that in _any_ partially observable (tabular) environment, committed Q-learning converges to feature values that represent a mixture of the "entrance state" values.
> Our key insight (Lemma 4.3) shows that in "rewire-robust" or "quasi-Markov" environments, a greedy policy with respect to these values is optimal.
> Nearly all reinforcement learning problems of practical relevance are either partially observable, or have a prohibitively large state space, requiring function approximation (such as state aggregation).
> Littman (1994) showed that finding a reactive policy in this setting is NP-hard in general.
> Thus, all efficient algorithms require further assumptions.
> To the best of our knowledge, this work introduces the weakest such assumption to date (rewire-robustness).
> We believe that this represents a significant and highly original contribution.
>
> **Limitations.**
> We discuss several limitations of our analysis, such as the restriction to stationary options, and our assumption of a finite set of options, and are happy to include any further limitations that you think deserve to be mentioned.
>
> We hope that we could address your concerns and answer your questions, and would be grateful about an increased score unless you have further concerns.

---

> > ### Author Rebuttal · Reviewer_yqeK · 2026-04-03
> >
> > Thank you for the explanation.  I now understand that the action is selected by a behavioural policy that is stochastic.  Furthermore, the algorithm converges to the optimal Q-function as opposed to the Q-function of the behavioural policy.  Hence, exploration can be ensured by the behavioural policy without impacting the convergence proof.  Thank you also for demonstrating that committed Q-learning can find an optimal policy in some environment where regular Q-learning completely fails.  My concerns are addressed and I am raising my score to weak accept.

---

### Decision · Program_Chairs · 2026-04-30

**Decision:**

Accept (regular)

**Comment:**

The paper proposes a novel algorithm called Committed Q-learning for POMDPs with deterministic observations. The authors prove that the algorithm converges to an optimal policy if the POMDP satisfies a property that they call rewire-robust. Essentially this means that a policy can be defined on the feature space and is invariant to the underlying state space. This is a weaker condition than the realizability condition often assumed for feature approximation.

The reviewers agree that the paper makes an important non-trivial theoretical contribution that increases the understanding of the type of POMDPs that can be solved efficiently. This is clearly of interest for the reinforcement learning community at large. For this reason the recommendation is to accept the paper for publication at ICML.